# Multi-centric origins and gene flow shape the diversity of β-thalassemia mutations in Southern East Asia

Qianqian Zhang[1,2,3,12], Jialong Li[2,3,12], Haoyang Huang[2,3], Xuan Shang[2,3], Yuhua Ye[2,3], Wei Zhang[2,3], Peng Lin[4], Yi Gong[5], Boon-Peng Hoh [6], Qingming Luo[1,4], Tizhen Yan[4], Xinghua Pan[1,7,8], Mark Stoneking [9,10], Shuhua Xu [11], Xiangmin Xu [2,3] ✉ & Lian Deng [11] ✉

Over 400 β-thalassemia mutations show population-differentiated spectra, yet their origins and evolution remain unclear. Focusing on targeted sequencing of 20,222 individuals and 510 β-thalassemia patients in southern China, we identified three major haplotype groups (HG) at the β-globin locus and observed highest haplotype diversity for CD41/42, -50, and HbE among 13 prevalent mutations in 993 carriers. Allele dating suggest these mutations emerged during agricultural expansions in the past 7420 years, represented by CD41/42 arising in mainland China. However, the -50 mutation likely originated on Hainan Island within 3900 years, subsequently spreading to the mainland and experiencing lineage-specific selection. HbE exhibits substantial haplotype heterogeneity in Yunnan, with network analyses indicating bidirectional disseminations between southern China and South/Southeast Asia. We further suggest an ameliorating effect of HG2, associated with elevated hemoglobin and fetal hemoglobin levels. These findings highlight multi-centric origins of β-thalassemia mutations and underscore the evolutionary context shaping their clinical impact.

β-Thalassemia is a common hereditary disease that demonstrates significant advantages for heterozygous genotypes in malaria-endemic regions, with an estimated 90 million carriers worldwide[1,2]. Over 400 different β-thalassemia mutations have been reported in the IthaGenes database (https://www.ithanet.eu/db/ithagenes/), a comprehensive and up-to-date database for hemoglobinopathies[3]. The β-thalassemia

mutation spectra vary across ethnic and geographical populations, each characterized by a few predominant mutations interspersed with rare mutations[4]. For instance, prevalent mutations include IVS-I-110 (*HBB*:c.93-21G > A), CD39 (*HBB*:c.118C > T), IVS-II-745 (*HBB*:c.316-106C > G), and IVS-I-6 (*HBB*:c.92+6T > C) in the Mediterranean populations, -29 (*HBB*:c.-79A > G) and -88 (*HBB*:c.-138C > T) in Africans,

[1]Dongguan Maternal and Child Health Care Hospital, Postdoctoral Innovation Practice Base of Southern Medical University, Dongguan, China. [2]Department of Medical Genetics, School of Basic Medical Sciences, Southern Medical University, Guangzhou, China. [3]Innovation Center for Diagnostics and Treatment of Thalassemia, Nanfang Hospital, Southern Medical University, Guangzhou, China. [4]Prenatal Diagnostic Center, Dongguan Maternal and Children Health Care Hospital, Dongguan, China. [5]Henan Provincial Key Laboratory of Genetic Diseases, Henan Provincial People's Hospital, People's Hospital of Zhengzhou University, Zhengzhou, China. [6]Division of Applied Biomedical Sciences and Biotechnology, School of Health Sciences, IMU University, Kuala Lumpur, Malaysia. [7]Precision Regenerative Medicine Research Centre, Medical Science Division, Macau University of Science and Technology, Macao, China. [8]Department of Biochemistry and Molecular Biology, School of Basic Medical Sciences, Southern Medical University, Guangzhou, China. [9]Department of Evolutionary Genetics, Max Planck Institute for Evolutionary Anthropology, Leipzig, Germany. [10]Biométrie et Biologie Évolutive, UMR 5558, CNRS & Université de Lyon, Lyon, France. [11]State Key Laboratory of Genetics and Development of Complex Phenotypes, Center for Evolutionary Biology, School of Life Sciences, Fudan University, Shanghai, China. [12]These authors contributed equally: Qianqian Zhang, Jialong Li. ✉e-mail: xixm@smu.edu.cn; denglian@fudan.edu.cn

and Hemoglobin E (HbE, *HBB*:c.79G > A), CD41/42 (*HBB*:c.126_129del), IVS-II-654 (*HBB*:c.316-197C > T), and CD17 (*HBB*:c.52A > T) in Southeast Asian populations[5–8]. β-Thalassemia exhibits significant regional disparity across China, with a pronounced north–south gradient in prevalence[9]. In northern China, only sporadic cases have been reported[10,11]. In southern China, where approximately 30 million of the total 270 million people are thalassemia carriers, different mutational spectra have been observed between the coastal and inland areas, and even across districts within a single province[8,12,13]. Moreover, the β-thalassemia mutational spectrum in southern Chinese populations differs substantially from that observed in Southeast Asian populations[11], underscoring the need for region-specific genetic characterization.

The origins and spread of β-thalassemia mutations have long been investigated. Considering the current distribution of β-thalassemia and the diversity of mutation patterns, it is unlikely that β-thalassemia originated from a single place and time and then spread throughout the rest of the malaria belt[14,15]. Complex origins and evolutionary scenarios for the β-thalassemia mutations have been comprehensively explored in the Mediterranean region and in West Asia[14]. However, β-thalassemia in southern East Asia has never been well characterized despite the high mutation prevalence, considerable ethnic diversity, and extensive epidemiological data in this region, all of which create an ideal setting for investigating the evolutionary history of β-thalassemia mutations.

Dissecting the haplotype background of disease mutations is crucial for understanding their origin and evolution. Previous studies have highlighted distinct haplotype structures and phenotypic consequences for a variety of trait-associated genes in East Asians, particularly in Chinese[16–18]. It has also been suggested that haplotype background could affect the clinical variations of hemoglobinopathies in African populations and in the mouse model[19,20]. Notably, recent haplotype analyses of the γ-globin gene region based on targeted long-read sequencing of 1020 β-thalassemia patients identified a haplotype associated with ameliorated symptoms[21].

Human demographic histories and adaptation have played a crucial role in shaping genetic susceptibility to diseases[22]. Thalassemia is one of the best-known examples of how natural selection acts on the human genome, as it may confer a reduced risk of malaria infection[14,15]. For example, the CD41/42 (*HBB*:c.126_129del) mutation, which is prevalent in the Chinese population, may be subject to natural selection and could also spread more widely through mechanisms such as gene conversion and migration[23]. A comprehensive investigation into the evolutionary history underlying the prevalence and distribution of the β-globin mutations could enhance our understanding of the underlying molecular mechanisms and provide additional insights into the phenotypic heterogeneity of the disease.

To trace the origin and evolution of β-thalassemia mutations in the Chinese, we re-analyzed next-generation sequencing data targeted on the β-globin locus from a previous study[12], comprising 20,222 Southern Chinese (SCN) individuals and 510 β-thalassemia patients from southern China, along with worldwide population genomic data, including 2504 individuals from the 1000 Genomes Project (KGP) Phase III and 929 individuals from the Human Genome Diversity Project (HGDP; Supplementary Fig. 1 and Supplementary Tables 1 and 2)[24,25]. We systematically characterized the haplotype structure of the β-globin locus in geographically and ethnically diverse Chinese populations, and then examined the haplotype phylogenies and estimated the age of the risk alleles. We further investigated the impact of the haplotype background on the clinical severity of β-thalassemia. We provide a workflow of this study in Supplementary Fig. 2 and an overview of the datasets and analytical tools in Supplementary Table 3.

## Results

### A global view of the haplotype structure at β-globin locus

The genetic structure of the SCN populations was assessed using targeted sequencing data spanning a total of 275.2 kb. A population phylogenetic tree was constructed based on pairwise population genetic distances as measured by $F_{ST}$, among the SCN populations and global populations from KGP and HGDP. The SCN populations generally clustered consistent with their geographic locations and ethnic affiliations (Supplementary Fig. 3). Linkage disequilibrium (LD) analysis indicated a strong LD block spanning 43.95 kb at the β-globin locus in the SCN samples (Fig. 1a and Supplementary Fig. 4). Extensive conservation of LD patterns was demonstrated among the predominant ethnic populations residing in southern China (Supplementary Fig. 5). The East Asians exhibited lowest nucleotide diversity (measured by $\theta_\pi$) when compared with the Europeans and Africans in the KGP dataset (Supplementary Fig. 6a). As expected, the African populations presented larger haplotype diversity (0.92–0.97), while the East Asian populations, except for JPT (0.79), generally displayed lower haplotype diversity (0.66–0.74), compared to other Eurasian populations (0.75–0.78) (Supplementary Fig. 6b). Interestingly, populations from the lower-latitude areas in East Asia (CHS, 0.72; CDX, 0.69; KHV, 0.66) had smaller haplotype diversities than those from the high latitudes (CHB, 0.74; JPT, 0.79). The haplotype diversity estimated for the SCN from the five provinces ranged from 0.64 to 0.72, consistent with that of East Asian populations in KGP (0.66–0.79).

The β-globin haplotypes represented two major haplotype groups in global populations (Fig. 1b). We identified 384 haplotypes in the block based on 44 SNPs with minor allele frequency (MAF) > 5% in SCN. Among these, 14 haplotypes (designated H1 to H14 in descending order of frequency in SCN) had frequencies >0.2%, collectively accounting for 94.7% of all haplotypes identified in SCN (Supplementary Table 4). The β-globin haplotype structure of the SCN populations closely resembled that of the East Asian populations from the KGP dataset, especially those from southern regions (e.g., CHS, CDX, and KHV) (Fig. 1c). Based on pairwise sequence differences measured by identity-by-state (IBS), we found that eight of the haplotypes were categorized in one major haplogroup (designated HG1), and the others could be further clustered into two haplogroups (designated HG2 and HG3, respectively) (Fig. 1a, b and Supplementary Fig. 7). The largest clade, HG1, diverged from the common ancestor of HG2 and HG3 approximately 606.8 thousand years ago (KYA; 95% confidence interval (CI): 592.6 KYA–617.1 KYA) assuming a fixed mutation rate of $1.25 \times 10^{-8}$ per base per generation. The lineages of HG1 could be traced to a common ancestor at around 340.7 KYA (95% CI: 318.1 KYA–365 KYA). H13 was isolated from other haplotypes in HG1, for which the time to the most recent common ancestor (TMRCA) was estimated to be 68.9 KYA (95% CI: 46.7 KYA–101.1 KYA). The divergence between HG2 and HG3 occurred at 306.7 KYA (95% CI: 292.7 KYA–323.8 KYA). Sub-lineages of these two haplogroups formed 232.7 KYA (95% CI: 94.5 KYA–366 KYA) and 46.8 KYA (95% CI: 43.9 KYA–49.3 KYA), respectively.

### Haplotype sharing of the β-thalassemia mutations

We focused on 13 β-thalassemia mutations that exhibit a high prevalence in southern China, accounting for 95% of all β-thalassemia cases in the region (Supplementary Figs. 8–9). The Human Genome Variation Society (HGVS) identifier and allele frequency (AF) of these 13 β-thalassemia mutations are listed in Supplementary Data 1. We identified 960 SCN individuals as carriers of one of these 13 mutations. Substantial heterogeneity was observed for the β-globin haplotypes carrying these mutations (Fig. 2a). We identified three general patterns of haplotype heterogeneity (Supplementary Tables 5–6 and Supplementary Fig. 10). Several mutations were linked to either a single haplotype or a limited number of haplotypes within specific haplogroups, leading to the low haplotype diversity (Fig. 2b). For example,

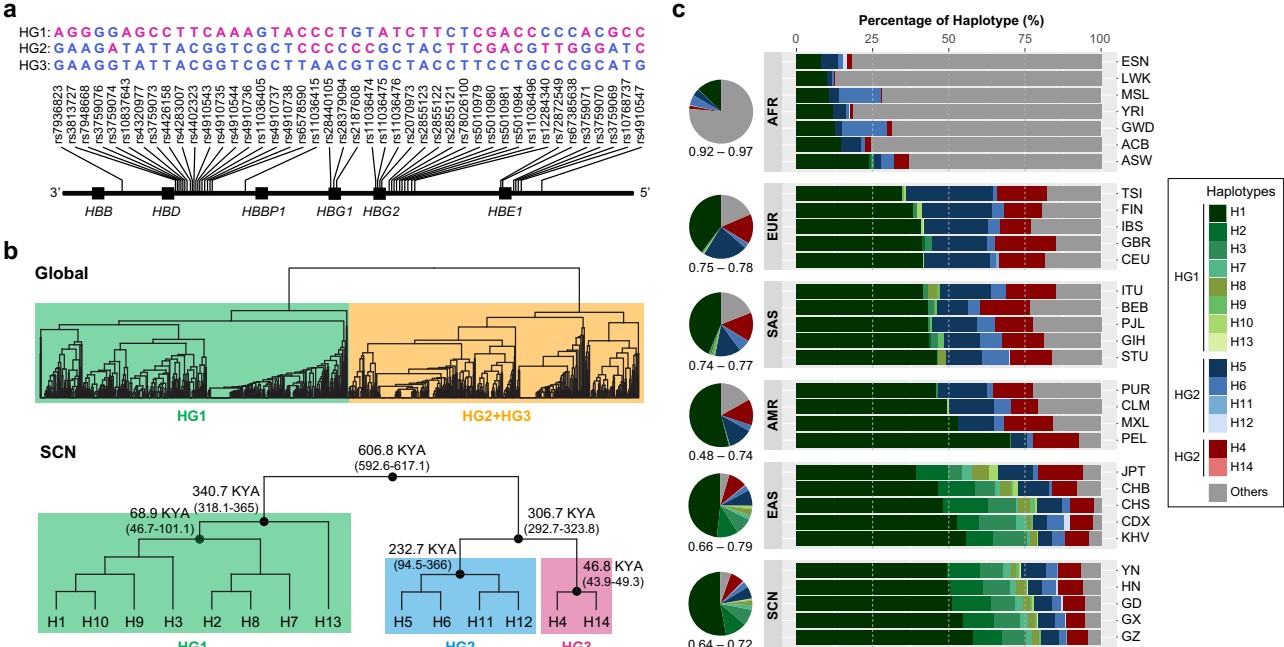

**Fig. 1 | Haplotype structure and diversity of the β-globin locus. a** Three major haplotype groups (HG1, HG2, and HG3) were determined based on 44 linked SNPs across the β-globin locus. Each group is represented by its dominant haplotype, with ancestral alleles shown in blue and derived alleles in red. A full list of haplotypes is provided in Supplementary Table 4. **b** Hierarchical clustering of the β-globin haplotypes in global populations from the 1000 Genomes Project Phase III, the Human Genome Diversity Project, and the Southern Chinese cohort (SCN) (upper), and that in the SCN populations (lower). In the SCN tree, the divergence time between two clades and the time to the most recent common ancestor across

all haplotypes within a clade are indicated on the branches (shown as mean values across 100 replicates, with 95% confidence intervals in brackets). **c** Distribution of β-globin haplotypes across global populations. Each horizontal bar represents the haplotype composition within a single population. Pie charts to the left show average haplotype proportions across populations within each region, with the corresponding range of haplotype diversity indicated below. Population abbreviations and additional details are provided in Supplementary Table 2. Source data are provided as a Source data file.

the codon insertions CD71/72 and CD27/28 were restricted to H1 and H6, respectively, while IVS-II-5, a very rare mutation (AF = 0.01%), presented with two haplotypes (H1 and H46) within HG1. Mutations such as -50, Hb NewYork, -29, and IVS-I-1 were shared by HG1 and one other haplogroup. The remaining mutations, including CD41/42, CD17, IVS-II-654, -28, and HbE with relatively higher frequencies (AF = 0.15%–1.04%) and CD43 with a very low frequency (AF = 0.02%), were found in all three haplogroups. We also noted that 90.8% of the -28 mutations were found in H14, a haplotype of HG3, while none of the other 12 β-thalassemia mutations were detected in H14. Additionally, 94% of H14 was linked to the -28 mutation (Supplementary Data 1 and Supplementary Table 5). This strong and exclusive association underscores a distinct connection between the -28 mutation and H14.

Although the overall haplotype structure at the β-globin locus was similar across populations and ethnic groups (Supplementary Fig. 11), we observed significant variation in the haplotype backgrounds of β-thalassemia mutations among provinces (Supplementary Figs. 12–14). We further investigated the four mutations with the highest haplotype diversity shown in Fig. 2a, including -50, HbE, CD43, and CD41/42. Intriguingly, the haplotype makeup of the -50 mutation on the Hainan Island involved a large proportion of HG3, markedly different from mainland populations (Fig. 2c). HbE displayed exceptionally high haplotype diversity and was shared across three haplogroups in Yunnan, while it was shared by two haplogroups in the other provinces. Most of the CD41/42 mutations were in HG1, while the proportions of the HG1 sub-lineages carrying this mutation varied widely across populations (Fig. 2c and Supplementary Fig. 14). The CD43 haplotypes were heterogenous in Guangxi and Guizhou but homogenous in Hainan, although the low allele frequency (AF < 0.02%) has hindered further investigation. Collectively, these observations suggested complex origins and evolutionary histories of β-globin mutations.

## A shared mainland origin of ethnicity-differentiated lineages of the CD41/42 mutation

The β-thalassemia mutations identified in the Hainan populations were less diverse than those in the other populations, with CD41/42 being most prevalent, constituting 64.9% of the total β-thalassemia mutations in Hainan (Supplementary Fig. 9). Particularly in the Li ethnic samples in Hainan (HN-Li), making up 20.4% of the total Hainan samples in our data, the allele frequencies of the CD41/42 mutation in total β-thalassemia mutations were 2.2–6.5 times higher than those observed in the Han ethnic group in Hainan (HN-Han) and other SCN ethnic populations (Fig. 3a). Even when compared with the Han Chinese (GX-Han) and Zhuang (GX-Zhuang) ethnic groups in Guangxi, which showed high genetic similarity with the Hainan populations at the β-globin locus ($F_{ST} = 0.0036$–0.0041; Supplementary Fig. 15), we still found the two ethnic populations in Hainan exhibited substantial differentiation in the β-thalassemia mutation spectrum.

Among the CD41/42 carriers in the HN-Li, the H2 haplotype was predominant over H1 (77.3% vs. 11.4%) (Fig. 3a). In contrast, the H1 haplotype was much more common (71.4%–77.5%) than H2 (16.9%–20.7%) in the GX-Zhuang, GX-Han, and other southern mainland Chinese populations. The HN-Han population presented intermediate haplotype frequencies between those of the HN-Li and mainland populations (H1: 38.5%; H2: 51.9%). Network analysis revealed a shared branch for the H1 and H2 haplotypes carrying the CD41/42 mutation (Fig. 3b). We observed lower nucleotide diversity on the haplotypes carrying the CD41/42 mutation in the HN-Li than in other populations (Supplementary Table 7). The allele age of the CD41/42 mutation in the HN-Li and GX-Zhuang was estimated to be 4.05 KYA–7.42 KYA (Supplementary Table 8), which aligns with their shared ancestral lineage tracing back to the ancient Bai-Yue people (4 KYA–11 KYA), who experienced a high incidence of malaria[26]. These

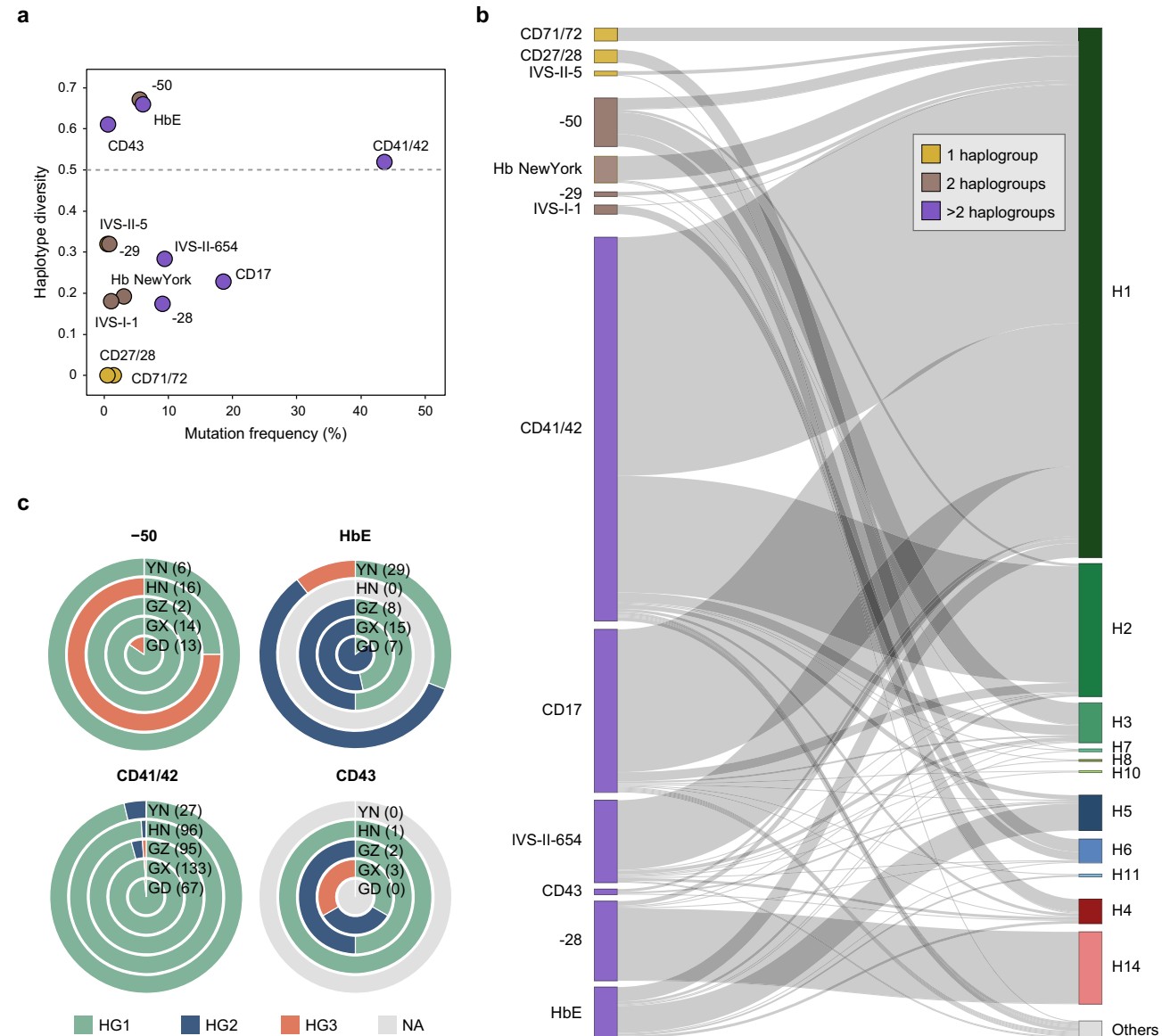

**Fig. 2 | Haplotype heterogeneity for the β-thalassemia mutations in southern Chinese. a** Frequencies and haplotype diversity of 13 β-thalassemia mutations among 960 carriers. The horizontal dashed line separates four mutations with markedly higher haplotype diversity. **b** Sankey diagram illustrating associations between β-thalassemia mutations (left) and β-globin haplotypes (right). Mutations linked to different numbers of haplogroups are color-coded, consistent with (**a**).

**c** Donut plots showing the distribution of haplogroups associated with four β-thalassemia mutations exhibiting the highest haplotype diversity, as indicated in (**a**). Data are presented for five provinces: Yunnan (YN), Hainan (HN), Guizhou (GZ), Guangxi (GX), and Guangdong (GD). The number of individuals carrying each mutation is indicated in brackets. Donut plots for the remaining nine mutations are shown in Supplementary Fig. 13. Source data are provided as a Source data file.

results suggested a shared origin of the CD41/42 mutation among the sub-lineages of Bai-Yue descendants, likely arising on the mainland. The observed enrichment of CD41/42-H2 in the HN-Li population may be attributed to a founder effect followed by genetic drift associated with subsequent isolation of the islanders.

### An island origin of the -50 mutation

Similarly, we observed large mainland-island haplotype differentiation at the -50 mutation (Fig. 3c), the second most prevalent β-thalassemia mutation in Hainan (12.8%) but much rarer (4.14%) among the β-thalassemia mutations in the mainland SCN populations. The -50 mutation linked to four haplotypes (e.g., H1, H2, H3, and H4) in the SCN populations (Supplementary Data 1). All four haplotypes belonged to HG1, and were observed in the mainland populations, with H1 (31.43%) and H3 (57.14%) being more common than H2 (5.71%) and H4 (5.71%). In the HN-Li population, however, the -50 mutation was exclusively

linked to H4 (100%). The haplotype network exhibited two deeply divergent clusters, HG1 and HG3, respectively, for the H1-3 and H4 lineages carrying the -50 mutation (Fig. 3d). The allele age for the -50 mutation was estimated to be 1.04 KYA–3.9 KYA in the SCN (Supplementary Table 8), postdating the split of HN-Li from the Bai-Yue ancestry on the mainland. We hypothesize an island origin of the -50 mutation in H4 in southern China. The -50 related haplotype and nucleotide diversity were higher in the HN-Han than in the mainland populations (Supplementary Table 7), with additional haplotypes carrying the -50 mutation exclusively observed in HN-Han (e.g., H13). Therefore, it is less likely that the enrichment of H4-linked -50 mutation on Hainan Island was caused by a founder effect during the island's settlement.

We then tested whether the -50 mutation arose from multiple independent events in the mainland and island populations (i.e., HG1 from the mainland and HG3 from the island). We constructed a local

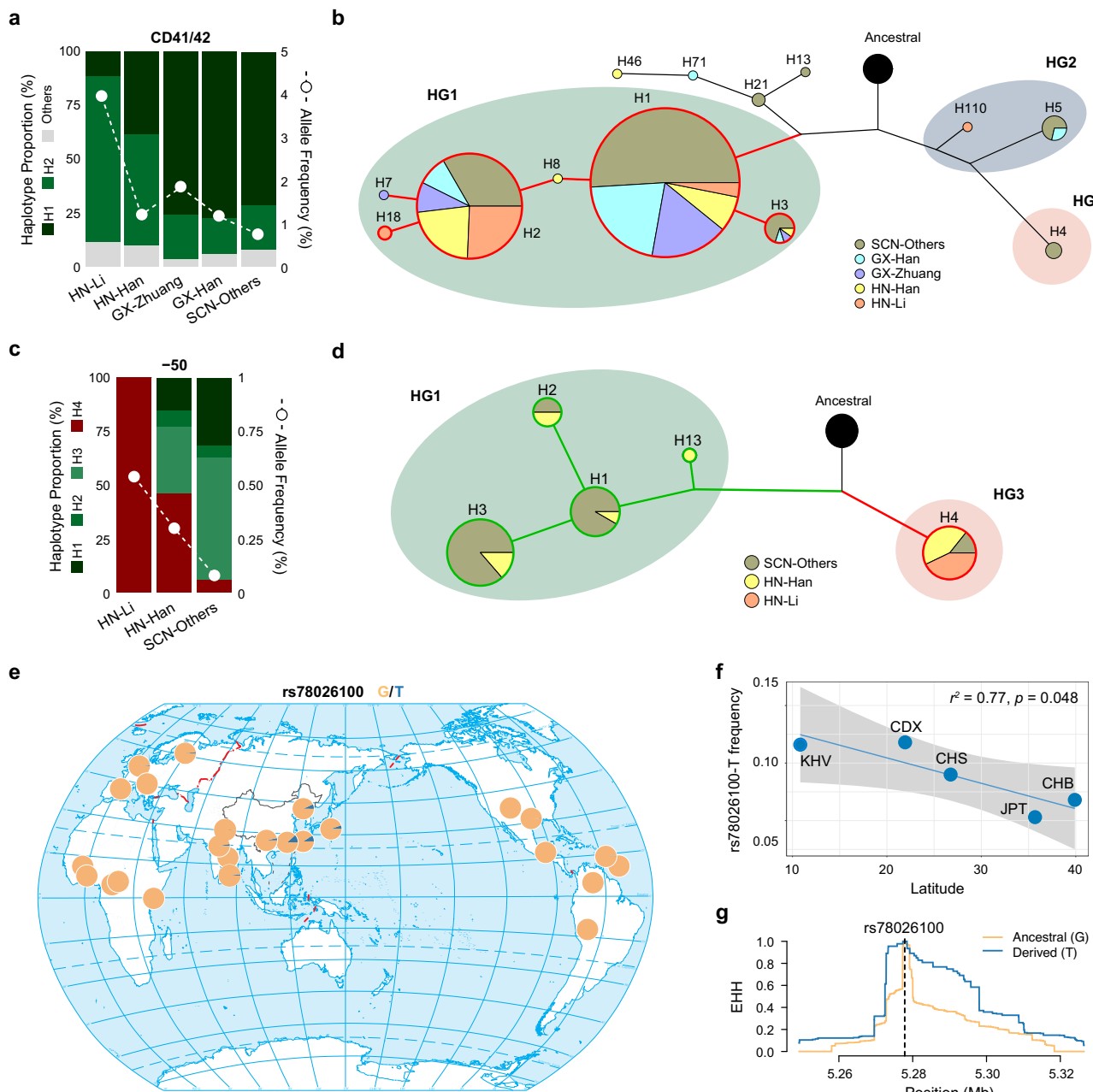

**Fig. 3 | Ethnic heterogeneity and haplotype network of the CD41/42 and -50 mutations across mainland and island populations. a** Allele frequencies and haplotype proportions of the CD41/42 mutation among Li and Han Chinese from Hainan (denoted as HN-Li and HN-Han, respectively), Zhuang and Han Chinese from Guangxi (denoted as GX-Zhuang and GX-Han, respectively), and ethnic populations from other provinces (denoted as SCN-Others). **b** Median-joining network of haplotypes carrying the CD41/42 mutation. The shared branch connecting the CD41/42-linked haplotypes (predominantly H1 and H2) is outlined in red. **c** Allele frequencies and haplotype proportions of the -50 mutation across ethnic populations from HN and other provinces. **d** Median-joining network of haplotypes carrying the -50 mutation. Distinct branches corresponding to the -50-linked haplotypes (HG1 and HG3) are highlighted in green and red, respectively. Networks in panels (**b**) and (**d**) are constructed based on the 44 linked SNPs.

**e** Global distribution of the derived allele T at rs78026100 (H3-specific) based on the 1000 Genomes Project Phase III data, indicating an East Asian-specific lineage. The map is adapted from the issued version (No. GS(2016)2962) of the Ministry of Natural Resources of China, downloaded from the Standard Map Service Website (http://bzdt.ch.mnr.gov.cn). **f** Positive correlation between latitude and rs78026100-T allele frequency in East Asian populations. A two-sided Pearson correlation test (single comparison; no multiple-testing correction applied) was performed: $r = -0.882$ (95% confidence interval (CI): -0.992 to -0.00075); $t = -3.249$ (degrees of freedom (df) = 3); $p = 0.0475$. Linear regression is shown for visualization: slope $\beta = -0.00221$ per degree latitude (95% CI: -0.00437 to -0.000045); $r^2 = 0.77$. The shaded area represents the 95% CI of the regression line. **g** Extended haplotype homozygosity (EHH) decay at rs78026100-T indicated selective sweep in East Asian populations. Source data are provided as a Source data file.

coalescent tree using RELATE for the SCN samples. All of the 13 β-thalassemia mutations examined were mapped onto the local genealogy, providing no strong evidence for recurrent mutations at these loci. HG1 was highly enriched in the East Asian populations (Kendall's tau-$b = 0.558$, $p = 0.001$), and the haplotype diversity within HG1 in East

Asian populations was 3.9–27.5 times greater than that in the other continental populations (Supplementary Fig. 16). There could have been a recent expansion of HG1 in the East Asian populations, as the TMRCA of the major haplotypes within HG1 was estimated to be 68.9 KYA, after the Out-of-Africa dispersal of the Eurasians. Interestingly,

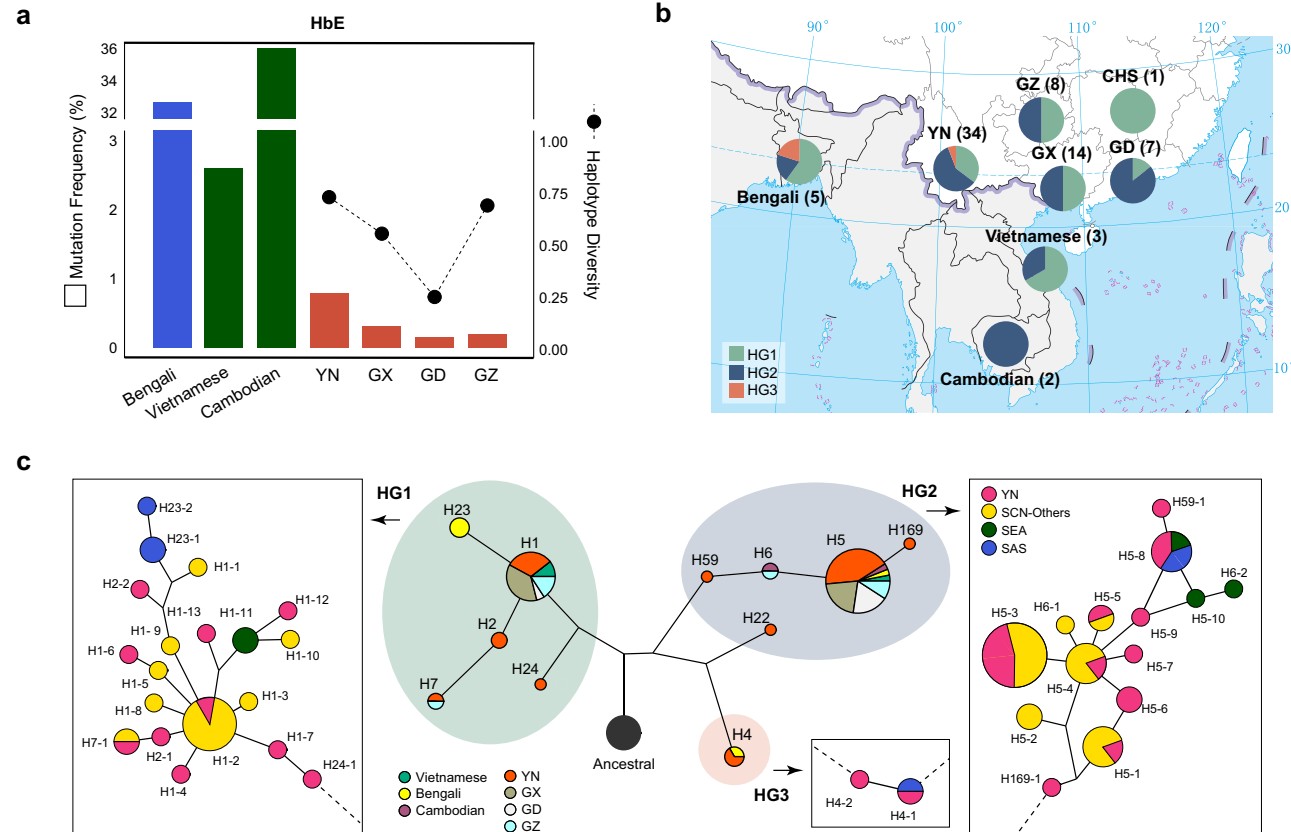

**Fig. 4 | Ethnic heterogeneity and haplotype network of the HbE mutation in Yunnan and surrounding regions. a** Prevalence of the HbE mutation in Yunnan (YN), Guangxi (GX), Guangdong (GD), and Guizhou (GZ), as estimated from our data, alongside reported frequencies for neighboring countries in South and Southeast Asia (e.g., Bangladesh, Vietnam, and Cambodia). No HbE carriers were identified in Hainan. Diversity of the HbE-linked haplotypes in the southern Chinese populations is indicated by the connected dots. **b** Haplotype frequency distribution of the HbE mutation in southern China, mainland Southeast Asia, and the northern Indian subcontinent. HbE carriers in Bengali, Vietnamese, and Cambodian are from the Human Genome Diversity Project (HGDP). One Dai individual from HGDP and two CDX individuals from the 1000 Genomes Project Phase III are included in the YN group. Sample sizes of each group are indicated in brackets. The map is adapted from the issued version (No. GS(2019)1669) of the Ministry of Natural Resources of China, downloaded from the Standard Map Service Website (http://bzdt.ch.mnr.gov.cn). **c** Median-joining networks of haplotypes carrying the HbE mutation based on 44 linked SNPs. Detailed sub-networks for each haplogroup, constructed from full-sequence data within the linked block, are shown in the boxed panels. Populations from different regions are color-coded. Source data are provided as a Source data file.

H3, which constituted 11.3% of HG1 in East Asians, was rarely observed in other populations: 0.6% in Africans, 0.8% in Americans, 1.5% in Europeans, and 2.8% in South Asians. H3 was characterized by an East Asian-specific variant rs78026100 (Fig. 3e). The variant is located in the noncoding region and has not been reported as a regulatory or trait-associated locus according to the GTEx Project (https://www.gtexportal.org/home/) and the GWAS catalog (https://www.ebi.ac.uk/gwas/). Nevertheless, signals of extended haplotype homozygosity (EHH), along with a significant latitude correlation, were observed at this locus in East Asian populations, suggesting possible recent adaptations (Fig. 3f, g). Based on these findings, we hypothesize that a recent selective sweep may account for the increased frequency of the -50 mutation in southern mainland China.

### Gene flow contributed to diverse lineages of the HbE mutation in Southwest China

HbE is the fifth most frequent β-thalassemia mutation in the SCN (Supplementary Data 1), with a notably high frequency and haplotype diversity in Yunnan Province, a region home to various ethnic groups (Fig. 4a and Supplementary Fig. 12). Our analyses revealed significant differences in the distribution of HbE haplotypes between populations from Yunnan and those from the surrounding areas (Fig. 4b). HbE was linked to the HG1 and/or HG2 haplotypes in populations from Guizhou, Guangxi, Guangdong, and Hainan, and those from the Indo-China

Peninsula in Southeast Asia (e.g., Cambodia and Vietnam). However, HG3, which was less prevalent in southern China (7.2%) and constituted 8.48% of the Yunnan samples, is notably more prevalent among HbE carriers in Yunnan. This non-negligible presence of HG3 also parallels the HbE haplotype structure in Bangladesh, which is located in the northeastern Indian subcontinent and geographically close to southwestern China. In the Indian subcontinent, the HbE mutation is restricted to the northeastern regions[27]. The HG3 haplogroup (predominantly H4 haplotype) exhibited a higher frequency in South Asians compared to East Asians, and notably, a higher proportion of HbE-associated haplotypes within HG3 was observed even with very limited samples. This suggested that gene flow from northern India or surrounding regions likely contributed to the HbE mutation diversity in Yunnan. The estimated allele age of the HG3-linked HbE mutation in Yunnan is 0.57 KYA–2.69 KYA (Supplementary Table 8), falling within the period of the Southwest Silk Road, which facilitated extensive human migrations between South Asia and China since the 2nd century BCE[28].

In contrast, an opposite pattern was observed for the HG1- and HG2-linked HbE mutation carriers. HG1 and HG2 were more prevalent in East and Southeast Asian populations than in South Asians (Fig. 1c). A haplotype network further revealed that the SCN samples carrying the HG1- or HG2-linked HbE mutation occupied hubs of the network, whereas their South and Southeast Asian counterparts appeared on

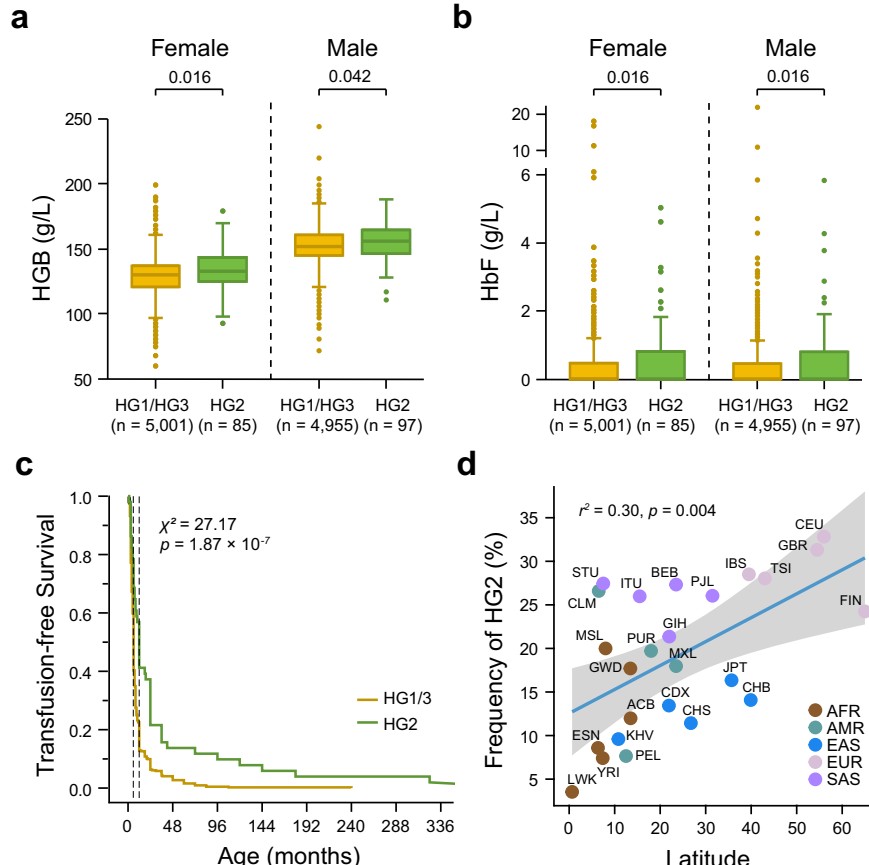

**Fig. 5 | Association between HG2 and thalassemia severity. a** Elevated hemoglobin (HGB) and **b** fetal hemoglobin (HbF) levels in HG2 homozygotes compared with HG1 and HG3 homozygotes, stratified by sex (5,052 males and 5,086 females). All individuals included in **a** and **b** do not carry β-thalassemia mutations and are homozygous for a single haplogroup. Sample sizes for each group are indicated in brackets. Box plots indicate median (middle line), 25th, 75th percentile (box), and 5th and 95th percentile (whiskers) as well as outliers (single points). For each trait, a two-sided Welch's t-test (assuming unequal variances) with Benjamini–Hochberg (BH) correction across the sex-stratified comparisons was applied, and the adjusted $p$ values are shown on the top of the boxes. In the HGB analysis, $t = -2.631$, degrees of freedom (df) = 86.09, and Hedges'g = -0.309 (95% confidence interval (CI): -0.55 to -0.10) for females; $t = -2.065$, df = 99.53, and Hedges'g = -0.215 (95% CI: -0.43 to -0.03) for males. In the HbF analysis, Welch $t = -2.568$, df = 85.04, and Hedges'g = -0.338

(95% CI: -0.54 to -0.11) for females; Welch $t = -2.842$, df = 97.19, and Hedges'g = -0.357 (95% CI: -0.54 to -0.16) for males. **c** Kaplan-Meier survival curves showing prolonged transfusion-free survival in patients with HG2 ($n = 51$) than those with other haplogroups ($n = 459$) in 510 transfusion-dependent β-thalassemia patients. The $p$ value was obtained using the log-rank (Mantel–Cox) test. **d** Positive correlation between the HG2 frequency and latitude across global populations from the 1000 Genomes Project Phase III dataset. Population abbreviations and additional details are provided in Supplementary Table 2. A two-sided Spearman correlation test (single comparison; no multiple-testing correction applied) was performed. Linear regression is shown for visualization: slope $\beta = 0.275$ percentage-points per degree latitude (95% CI: 0.104 to 0.446); $t = 3.337$, degrees of freedom (df) = 22; $F(1,22) = 11.137$. The shaded area represents the 95% CI of the regression line. Source data are provided as a Source data file.

the peripheral branches (Fig. 4c and Supplementary Fig. 17). This pattern suggests that SCN may represent a source population of the HbE mutation, which subsequently spread to South and Southeast Asian populations through genetic interactions. Supporting this hypothesis, a recent ancient genome study reported the dispersal of diverse ancestries from Yunnan to present-day Austroasiatic speakers in South and Southeast Asia since approximately 5.5 KYA[29].

## Ameliorating effects of HG2 on hematological indicators

We further investigated whether the haplotype background of β-globin mutations may influence the hematological phenotypes. We analyzed 10,138 (5052 males and 5086 females) SCN individuals carrying $\beta^N/\beta^N$ and who were homozygous for β-globin haplogroups. Hematological indices were compared across haplogroups separately for males and females. Our analysis revealed that HG2 was significantly associated with higher levels of hemoglobin (HGB) and fetal hemoglobin (HbF) (Fig. 5a, b). We then replicated this analysis in a cohort of 510 β-thalassemia patients with strictly controlled globin genotypes

($\alpha\alpha/\alpha\alpha$ and $\beta^0/\beta^0$). Consistently, we found that HG2 was also significantly associated with prolonged transfusion-free survival ($p < 0.0001$; Fig. 5c), suggesting a potential ameliorating effect of this haplogroup. Interestingly, the HG2 frequency across global populations showed a positive correlation with latitude ($r^2 = 0.30$, $p = 0.004$; Fig. 5d), with lower frequencies observed in regions of high thalassemia prevalence. This pattern was also evident within East Asia (9.6% in KHV, 11.4% in CHS, 14.1% in CHB, and 16.3% in JPT). In contrast, HG2 was generally less frequent in Africa (median frequency: 9%) but most prevalent in European populations, where frequencies ranged from 24.2% to 32.8% (Fig. 5d).

To explore potential functional mechanisms, we identified seven variants specific to HG2 and absent in other haplogroups (Supplementary Fig. 18a). Single-variant association analyses in β-thalassemia patients revealed that these HG2-specific variants were significantly associated with elevated HbF levels (Supplementary Fig. 18b). Notably, epistatic analyses demonstrated that five of these variants exerted a synergistic effect on HbF expression when interacting with rs7482144, an Xmn-1 polymorphism in the $\gamma^G$-globin gene *HBG2* that is a well-

established quantitative trait locus for HbF levels (Supplementary Fig. 18b)[30].

## Discussion

This study presents a comprehensive investigation of the genomic landscape underlying haplotype diversity and the evolutionary dynamics of β-thalassemia mutations in southern East Asia, with a particular focus on the ethnic-specific haplotype structures that reflect how population migration and admixture have shaped the genetic architecture. Although some homologous regions within the β-globin locus may pose challenges for genotyping, the high-density next-generation sequencing data provided us with sufficient coverage depth (178×–191× on average for the targeted regions), ensuring the accuracy of genotype determination. Notably, 97.9%–98.6% of the nucleotides were covered by >20 uniquely mapped reads, supporting reliable variant calling, even for the β-thalassemia mutations with rare frequencies[12]. Our analyses of haplotype structure revealed intricate associations between the β-globin haplotypes and β-thalassemia mutations, as well as non-random patterns of differentiation among ethnic populations. Additional evidence from haplotype diversity, phylogenetic analysis, allele age estimation, and historical records indicates possible scenarios for the origin and dissemination of the highly variable β-thalassemia mutations. Previous studies suggested a single origin of the sickle hemoglobin (HbS) mutation (β[S]) in central Africa during the late Pleistocene to early Holocene, followed by geographic spread and diversification associated with the Bantu expansions (Fig. 6a)[31,32]. Similarly, our analyses did not support multiple independent origins for each single β-thalassemia mutation among southern Chinese populations, even when focusing on four mutations (e.g., CD41/42, -50, HbE, and CD43) with the highest haplotype diversity among the 13 examined. Nevertheless, the broader landscape encompassing variable β-thalassemia mutations reveals more complex scenarios than those documented for the HbS mutation in Africa, likely driven by the rich geographic and ethnic diversity of southern China (Fig. 6b).

It has been widely acknowledged that the carrier-resistance to malaria conferred by the β-thalassemia mutations has resulted in the local amplification of β-thalassemia in the malaria-endemic regions along the equatorial belt[33,34]. This process has also been associated with two major transitions of modern human living environments and subsistence, the out-of-Africa dispersal and agricultural expansions (Fig. 6a). *Plasmodium falciparum* (*P. falciparum*), the most lethal malaria parasite responsible for 90% of malaria-related mortality worldwide, and *Plasmodium vivax* (*P. vivax*), the most prevalent species particularly in non-African regions, are both believed to have originated in Africa and were likely introduced to Asia during the expansion of Eurasian populations around 50,000–60,000 years ago[35–37]. However, historical records of malaria outbreaks emerged only after the advent of agriculture. The earliest documented cases of fevers potentially attributable to malaria in China date to ~5000 years ago, likely caused by *P. vivax*[38,39]. While we do not exclude the possibility of substantial malaria endemicity prior to that time, the transition to irrigated wet-rice agriculture in southern East Asia beginning around 6000 years ago likely facilitated the proliferation of malaria vectors, thereby intensifying the selective pressure for thalassemia mutations in local populations[40,41]. Our data indicate that the major β-thalassemia mutations in SCN populations likely originated in Asia within the past 8000 years. This estimated time aligns with the expansion of agriculture in southern East Asia and the adjacent Southeast Asian regions.

Despite the linguistic and cultural diversity, most Hainan populations trace their ancestries to southern mainland China[42]. The HN-Li people, descended from the ancient Bai-Yue, were thought to be the earliest settlers and have been residing on the island for over 4000 years[26]. The Han Chinese migration to Hainan began during the Qin Dynasty (~2200 years ago), with major influxes in the 16th and 17th centuries from Fujian and Guangdong[43]. We observed mainland-island differentiation on the haplotype structures at the CD41/42 and -50 mutations. The HN-Li population exhibited more homogenous haplotypes, predominantly H2 for the CD41/42 mutation and H4 for the -50 mutation, whereas the mainland SCN populations mainly carried H1 and H3 for these two mutations, respectively. The HN-Han population displayed intermediate haplotype frequencies between HN-Li and other mainland SCN populations (Fig. 3a, c). Notably, the predominant haplotypes associated with the CD41/42 mutation (H2 in HN-Li and H1 in mainland populations) belonged to the same haplogroup (HG1). In contrast, those linked to the -50 mutation (H4 in HN-Li and H3 in mainland populations) belonged to two deeply divergent haplotype groups, HG3 and HG1, respectively (Fig. 3b, d). The HN-Li carriers of either CD41/42 or the -50 mutation exhibited lower nucleotide and haplotype diversity, which could, in principle, suggest a mainland origin followed by founder effects and genetic drift within the isolated island population. However, we propose a more likely island origin for the -50 mutation for the following reasons. First, the Hainan population, including HN-Li and HN-Han, exhibited substantially higher haplotype diversity associated with the -50 mutation compared to mainland SCN populations. Second, the estimated allele age of 1.04 KYA–3.9 KYA supports a post-settlement origin of the -50 mutation on Hainan Island. Notably in HN-Li, although the -50 mutation showed a markedly higher allele frequency than in other SCN populations, it was exclusively linked to haplotype H4, which showed comparable frequencies across most SCN populations. This pattern may reflect a recent origin of the H4-linked -50 mutation or the effect of genetic isolation on the HN-Li population. However, while individual ethnic identity in this cohort was determined through self-reporting and verified by household registration, the possibility of mismatches between self-reported and genetic ancestry cannot be entirely excluded. Although the overall cohort size was large, we acknowledge that the observed frequency of the H4-linked -50 mutation may reflect sampling bias due to the limited number of mutation carriers. Despite these unavoidable confounding reasons, the enrichment of the H4-linked -50 mutation in Hainan Island remains a robust observation.

The HbE carrier rate was significantly lower in China (0.3%) than has been reported in northeastern India (64.5%), Bangladesh (32.78%), and mainland Southeast Asia (Thailand 22.3%, Myanmar 10.1%, Cambodia 35.9%)[44–47]. Analysis of Southeast Asian populations suggests that the HbE mutation in Thailand originated 1.24 KYA–4.4 KYA[48], and multiple independent origins were also suggested for the HbE mutation in Southeast Asia[49–51]. A study proposed that an HbE mutation found in Yunnan shares the same origin as one found in Thailand, and also indicated that consanguineous marriage might explain the variation in the frequency of HbE[52]. The HbE mutation found in northeast India may have been introduced through Austroasiatic migrations from Southeast Asia, or it may have originated in India and subsequently spread through human population movements, particularly from India to Southeast Asia[44]. In this study, we propose possible dispersals of the HbE mutation from southern China to South and Southeast Asia based on the observed haplotype diversity and genealogical structure of HG1 and HG2 haplotypes carrying the HbE mutation. Yunnan has served as a hub of the Southern Silk Road since around 2000 years ago, connecting central China with Southeast Asia and extended to India and beyond[28]. It was also central to the Ancient Tea Horse Road, a trade network that began approximately 1500 years ago that linked the tea-producing regions of southwest Yunnan with markets in Tibet, Southeast Asia, Nepal, and central China[53]. Genetic studies have revealed shared ancestries and extensive gene flow among southern China, mainland Southeast Asia, and South Asia[54–57]. In particular, recent ancient genome evidence suggests that Yunnan may have been a source region for the Austroasiatic expansion into Southeast Asia and northeast India within the past 5500 years, which

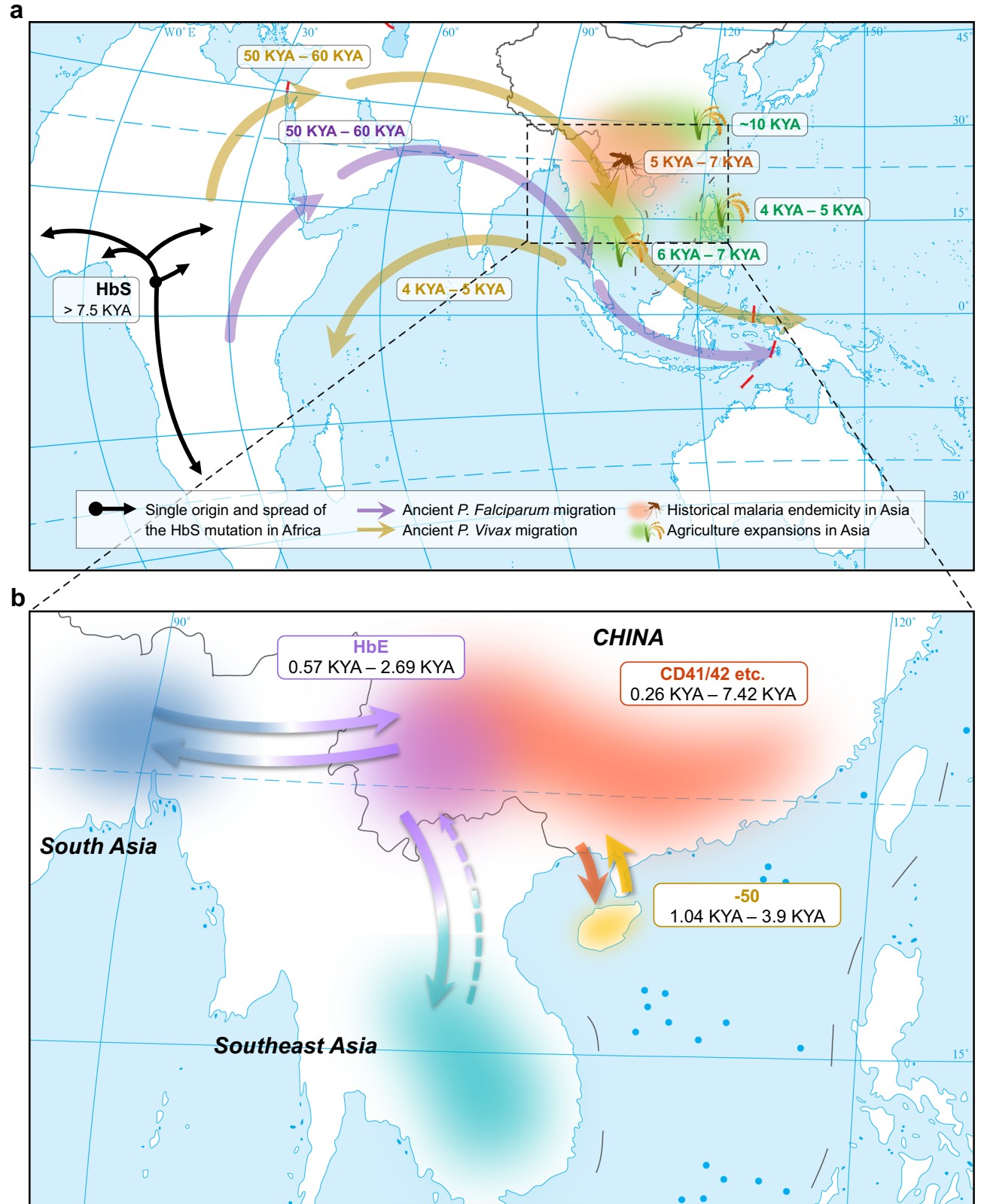

**Fig. 6 | Landscape of the origin and evolution of β-thalassemia mutations in southern East Asia. a** Global context of the origin and evolution of thalassemia, based on the literature. The map illustrates ancient migrations of two major malaria parasites (*Plasmodium falciparum* and *Plasmodium vivax*) from Africa to Asia, historical malaria endemicity in China, as well as early agricultural expansions around southern China. For comparison, the hypothesized single origin of the HbS mutation in Africans is also shown. **b** Proposed multi-centric model for the origin and migration of β-thalassemia mutations in southern East Asia. Solid arrows indicate spread routes inferred from the current study, while the dashed arrow represents migration reported in the literature. The map is adapted from the issued version (No. GS(2016)2962) of the Ministry of Natural Resources of China, downloaded from the Standard Map Service Website (http://bzdt.ch.mnr.gov.cn).

coincides with the period of historically reported malaria endemicity in China[29].

The other β-thalassemia mutations did not exhibit as high haplotype diversities as those seen in the CD41/42, -50, or HbE mutations, and they were generally more prevalent in the mainland populations. For instance, the CD17 mutation, which accounted for approximately 20% of β-thalassemia cases in mainland China, was almost absent on Hainan Island. Likewise, the -28 and IVS-II-654 mutations were rare or absent in the HN-Li population but presented in the HN-Han population. These patterns suggest that these mutations likely originated in mainland China after the initial settlement of the Li people and were later introduced to Hainan through subsequent Han Chinese migration. Allele age estimates further supported this hypothesis (Supplementary Table 8). Taken together, our results suggest multi-centric origins of multiple β-thalassemia mutations, with subsequent population genetic interactions contributing to their current prevalence and diversity.

Haplotype analysis enables a re-examination of the phenotypic associations of β-thalassemia mutations within their evolutionary context. Early evidence emerged from studies on small groups of Africans with sickle cell anemia, where distinct β-globin locus haplotypes were linked to varying hematological traits[58]. A recent study using targeted long-read sequencing of β-thalassemia patients reconstructed a 7.1 kb haplotype of the γ-globin gene region and identified a specific haplotype associated with disease phenotypes[21]. In the current study, our data support the classification of β-globin haplotypes spanning 44 kb in SCN populations into three groups, which might be associated with variation in clinical phenotype or disease severity. Notably, we found that HG2 was strongly associated with elevated HbF and hemoglobin levels (Fig. 5a, b). However, downstream analyses of HG2-specific variants suggest that its effect may not be fully explained by canonical cis- or trans-regulatory mechanisms. The identified variants do not reside within well-characterized enhancers of the globin genes, and only a subset, such as rs72872549, shows limited bioinformatic evidence of potential interaction with transcription factors like KLF1 (Supplementary Fig. 18c and Supplementary Table 9). These findings raise the possibility that the regulatory effects of HG2 may involve more complex mechanisms, such as higher-order chromatin architecture or epigenetic priming, which would require further functional studies to elucidate. Beyond potential functional consequences, the observed latitudinal decline in HG2 frequency across Chinese and global populations raises intriguing questions about possible evolutionary pressures. Although malaria-driven selection is a well-established force shaping β-thalassemia mutations, a direct connection between HG2 and malaria resistance remains speculative. Previous studies have highlighted complex interactions between HbF levels and malaria susceptibility[59,60], but the specific evolutionary forces underlying the HG2 distribution remain unclear and merit further investigation.

There are some important limitations to our study. The human β-globin locus is subject to complex natural selection, including both balancing and positive selection. These opposing forces on allele fixation may obscure the accurate determination of the underlying selection regime. An outlier strategy that compares the β-globin gene with other neutral genomic regions could help elucidate the adaptive evolution of this gene across populations. However, this approach is not possible with the targeted sequencing data used in this study and should be the subject of future study. Natural selection may also introduce potential bias in the allele age estimation in our analyses. Furthermore, current approaches for allele age estimation, including RELATE, Genealogical Estimator of Variant Age, and Runtc etc., provide reliable estimates of allele age from whole-genome datasets[61-63]. We selected RELATE for this analysis because it outperformed other estimators by, at least partially, mitigating the issue of haplotype tracks terminating in the intergenic or intronic regions[64]. Nevertheless, the targeted sequencing data used in our study may still result in an underestimation of the allele age (Supplementary Fig. 19). Alternatively, we sought to use ancient human genomes to trace the allele dynamics throughout human history. However, most of the β-thalassemia mutations (except the HbE mutation, which, however, had no identified carriers in the ancient samples) and the 44 SNPs (except five SNPs) we used to construct the β-globin haplotypes, were not successfully genotyped in the latest release of ancient genomes from the Allen Ancient DNA Resource project[64]. In addition, several modifier genes and variants reported to have large impacts on the clinical heterogeneity of β-thalassemia are on other chromosomes and hence have not been fully captured in this study[65]. However, given that the inter-chromosomal associations are less common, it is reasonable to expect a low degree of linkage between the β-globin haplotypes and the genotypes of the modifier variants. Despite these limitations, our study emphasizes the importance of a detailed investigation of haplotype structure and evolution in understanding their impact on disease phenotypes. Future studies incorporating whole-genome sequencing data from diverse ancestries, along with more advanced approaches for phasing the rare mutations, will enhance our understanding of the genetic architecture and evolutionary mechanisms underlying β-thalassemia.

## Methods
### Data collection and processing
Targeted sequencing data for the β-globin locus region were obtained from two cohorts in a previous study[12]. The first cohort included 20,222 randomly collected samples from five provinces of southern China (Supplementary Fig. 1): Guangxi (GX, $n = 4834$), Guangdong (GD, $n = 4622$), Yunnan (YN, $n = 3964$), Guizhou (GZ, $n = 4082$), and Hainan (HN, $n = 2720$). Among these 20,222 samples, 960 were identified as positive carriers, each carrying one of the 13 β-thalassemia mutations. The second cohort consisted of 510 β-thalassemia patients with the $\beta^0/\beta^0$ genotype from southern China. Targeted sequencing was conducted using 100-bp paired-end reads on Illumina HiSeq2000 or HiSeq4000 platforms, achieving an average sequencing depth of 178×–191× in the target region. The targeted β-globin locus region (chr11:5,246,604–5,312,691, GRCh37) included five protein-coding genes (HBB, HBD, HBG1, HBG2, and HBE1) and several erythroid-specific enhancer elements (HS-1, HS-2, HS-3, HS-4, and HS-5). A total of 1,594,170 variants were detected and annotated in this genomic region. The ancestral allele at each position was determined based on the ancestral sequence released by KGP (https://www.internationalgenome.org/category/phase-3/)[24]. Haplotype phasing was performed using Eagle version 2.4.1 (https://data.broadinstitute.org/alkesgroup/Eagle/) in a reference-free mode[65]. Missing genotypes were imputed using the -noImpMissing parameter, with all other settings left at their default values. Self-reported ethnicity of the samples was collected through standardized questionnaires and verified using local household registries.

Genome data of global populations released by the KGP (30×; https://www.internationalgenome.org/) and the HGDP (10–30×; http://www.hagsc.org/hgdp/) were also analyzed in this study[24,25]. Genotypes within the target region were extracted and analyzed, comprising 2418 variants from 2504 KGP samples and 2005 variants from 929 HGDP samples. Detailed information on the populations and the analytical tools used are provided in Supplementary Tables 2 and 3, respectively. Adobe Illustrator version 26.5 (https://www.adobe.com/products/illustrator.html/) and PowerPoint for Mac Version 16.16.27 (https://www.microsoft.com/) were used to design graphical elements such as shapes, icons, and lines, as well as assemble and organize the figures.

## Statistical and population genetic analyses

**Population differentiation and genetic structure.** We assessed the population genetic structure using these targeted sequencing data. We calculated pairwise $F_{ST}$ values between populations using the Weir & Cockerham estimator, which is implemented in VCFtools version 0.1.14 (http://vcftools.sourceforge.net/), to assess genetic differentiation[66,67]. The population phylogeny was constructed based on the $F_{ST}$ matrix using the unweighted pair group method with arithmetic mean (UPGMA) using the hclust function implemented in the stats package of R version 4.3.1 (https://www.r-project.org/).

**Defining the LD block and dissecting the haplotype structure.** LD was estimated for SCN samples using PLINK version 1.9 (https://www.cog-genomics.org/plink2/) and Haploview version 4.2 (https://www.broadinstitute.org/haploview/haploview)[68,69]. The haplotype blocks were determined using multiple approaches implemented in Haploview, including the four-gamete rule, confidence intervals of D', and the solid spine of LD. Adjacent haplotype blocks with linkages > 0.8 were merged, and we focused on the major block spanning 43.95 kb (chr11:5,250,168–5,294,120). Primary haplotypes within this block were determined based on 44 SNPs that satisfied Hardy–Weinberg equilibrium ($p > 0.001$) and had a minor allele frequency greater than 0.05.

We used an in-house script to measure the haplotype diversity ($H_d$) at the β-globin locus for each population, which was calculated as follows:

$$H_d = 1 - \sum p_i^2$$

where $p_i$ is the estimated frequency of the $i^{th}$ haplotype. An IBS distance matrix for all haplotypes was constructed using PLINK version 1.9[68]. Hierarchical clustering analysis based on the haplotype IBS matrices was conducted using UPGMA using the hclust function implemented in the stats package of R version 4.3.1 (https://www.r-project.org/).

**Constructing the haplotype network.** VCF files were first converted to FASTA files using an in-house script, and then NEXUS formatted files were generated using DnaSP6 version 6.12.03 (http://www.ub.edu/dnasp/)[70]. The NEXUS format is compatible with various software programs for phylogenetic analysis. These NEXUS files were then input into PopART version 1.7 (https://popart.otago.ac.nz)[71], where the haplotype network was constructed using the median-joining method to explore the phylogenetic relationships of the β-globin haplotypes.

**Estimating the TMRCA and the divergence time.** We estimated the TMRCA within each haplogroup and the divergence time between haplogroups using 146 informative genotypes from the haplotype block. A total of 100 replicates were performed, with 25 sequences randomly sampled from each group per replicate. For both TMRCA and divergence time, the mean values were calculated across replicates, and 95% confidence intervals were derived empirically from the 2.5th and 97.5th percentiles of their respective distributions. This analysis was conducted using a TMRCA calculator available at https://github.com/Shuhua-Group/TMRCA/, assuming a divergence time of 6.5 million years between humans and chimpanzees.

**Testing for neutrality.** Natural selection analyses were conducted for the East Asian populations using the KGP data. We performed rehh version 2.0 (https://cran.r-project.org/web/packages/rehh/index.html) to estimate the EHH[72,73]. We also estimated the nucleotide diversity ($\theta_\pi$) across different populations using a public tool available at https://github.com/Shuhua-Group/Theta_D_H.Est/, with a sliding window of 50 kb and a step size of 10 kb.

**Estimation of the allele age.** Allele age was estimated using RELATE software version 1.2.2 (https://myersgroup.github.io/relate/)[61]. We first converted the phased VCF files into .hap and .sample files using the RelateFileFormats tool. These files were then refined using the PrepareInputFiles.sh script, incorporating the ancestral sequence and genome mask files as recommended in the RELATE user manual. For allele age estimation, we adjusted the effective population size to reflect population-specific differences while using the default mutation rate of $1.25 \times 10^{-8}$ per base per generation. The analysis yielded estimated ages (in generations) for both the lower and upper boundaries of the allele branch lengths, which were subsequently converted to years by assuming a generation time of 25 years.

**Haplotype-based association analysis.** Hematological indicators were compared between haplogroups using the Mann-Whitney U test. Transfusion-free survival curves were generated using the Kaplan-Meier method and compared by the log-rank test. Pearson's correlation test was applied to assess the relationship between the haplogroup frequency and geographic latitude across populations. To evaluate the association between haplogroup frequency and population groupings within East Asia, we applied the Kendall's tau-b test. All statistical analyses were performed using SPSS version 22.0 (http://www-01.ibm.com/software/analytics/spss/). Graphs were generated using GraphPad Prism version 9.4.1 (https://www.graphpad.com/) and R version 4.3.1 (https://www.r-project.org/).

**Single-SNP association and epistasis analysis.** We evaluated the association between seven HG2-specific SNPs and HbF levels using PLINK version 1.9 (https://www.cog-genomics.org/plink2/)[68]. As a positive control, we selected a well-established HbF-associated SNP, rs7482144, which is located 158-bp upstream to *HBG2* (Xmn1 C-T restriction site). Additionally, the interactive effects of rs7482144 and the seven HG2-specific SNPs on HbF levels were tested using the epistasis option in PLINK version 1.9.

## Ethics statement

The sequencing data for Chinese populations were sourced from a previous study[12], with ethical approval (NFEC-2015-03) granted by the Medical Ethics Committee of NanFang Hospital. All procedures were in accordance with the ethical standards of the Responsible Committee on Human Experimentation and the Helsinki Declaration of 1975, as revised in 2000.

## Reporting summary

Further information on research design is available in the Nature Portfolio Reporting Summary linked to this article.

## Data availability

The genomic variation data of the southern Chinese populations analyzed in this study have been deposited in the Genome Variation Map at the National Genomics Data Center, China National Center for Bioinformation/Beijing Institute of Genomics, Chinese Academy of Sciences, under the accession number GVM000968. Due to national policy restrictions on human genomic data and approval requirements from the Ministry of Science and Technology of the People's Republic of China, only controlled access is permitted. Data access can be requested via the accession number in the repository or by contacting the corresponding author (Xiangmin Xu, xixm@smu.edu.cn). There are no specific restrictions on applicants, and requests for scientific purposes will, in principle, be approved within two weeks following an internal review. Once access is granted, at least four weeks will be allowed for data download. All processed data supporting this study are provided with the paper, including the Supplementary Information and Source Data files. The processed data and code are also available at https://codeocean.com/capsule/

9987989/tree. Source data are provided with this paper. Data release has been approved by the Ministry of Science and Technology of the People's Republic of China (2025BAT00488). Source data are provided with this paper.

## Code availability

Computer code is available at https://codeocean.com/capsule/9987989/tree.

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

## Acknowledgements

We acknowledge support by research funding from National Natural Science Foundation of China (82402163 to Q.Z.; 82471895 to X.X.; 82202044 to Y.G.; 32270665 to L.D.; 32288101 and 32030020 to S.X.), Natural Science Foundation of Hainan Province (823QN349 to Y.G.), National Key Research and Development Program of China (2023YFC2605400 to S.X.; 2022YFC3400300 to L.D.), Shanghai Science and Technology Commission Program (25JS2810100 & 23JS1410100 to S.X.), and Guangdong Medical Research Fund Project (A2024305 to T.Y). The funders had no role in the study design, data collection, analysis, decision to publish, or preparation of the manuscript.

## Author contributions

X.S., Y.Y., and X.X. provided research data; Q.Z. and L.D. conceived the study; Q.Z. and J.L. analyzed the data; L.D., Q.Z., and J.L. interpreted the analyses; Q.Z., H.H., W.Z., and P.L. drafted the manuscript; L.D., S.X., Y.G., T.Y., X.P., B.P.H., M.S., Q.L., X.S. and Y.Y. revised the manuscript; L.D. and X.X. supervised the project.

## Competing interests

The authors declare no competing interests.
