## [Transparent Peer Review file · Nature Communications]

Multi-Centric Origins and Gene Flow Shape the Diversity of β -Thalassemia Mutations in Southern East Asia

Corresponding Author: Professor Xiangmin Xu

Version 0:

Reviewer comments:

Reviewer #1

(Remarks to the Author)

In this manuscript, Zhang and colleagues re-analyzed targeted next-generation DNA sequencing data of the beta-globin haplotype to infer the genetic history of beta-thalassemia mutations in Southern China. The beta-globin haplotype is interesting because it has been continuously under natural (positive and balancing) selection throughout human evolution. In China, like in other parts of the World where malaria is (or was) endemic, mutations arose that confer a protective advantage when heterozygous, but are detrimental in the homozygous states. In their study, the authors describe how allele, haplotype and haplogroups frequencies vary in Southern China, and proposed several hypotheses (e.g. founder effect in Hainan, the Silk Road for the HbE mutation) that may explain these fluctuations. Finally, they showed how the beta-globin haplotypes (and polymorphisms within) associate with hemoglobin levels in healthy participants and blood transfusion in beta-thalassemia patients.

This is a re-analysis of existing data. While the phylogeny analyses are robust (given the local genetic data available), the conclusions are not particularly novel. I am also not very convinced by their attempt to annotate in silico one of their top variants. I have the following comments:

1. I think that it should be made clear in the abstract and at the beginning of the Results section that this is not a whole-genome sequence project (this was my initial impression).
2. Given the lack of genome-wide genetic data, how did the authors ascertain ethnicity or genetic structure? Is that self-reported? How accurate are these self-reports? Would 5-10% inaccuracies (for instance for the residents of Hainan) impact your main conclusions?
3. While it was originally reported that the sickle cell mutation (HbS) arose independently on 5 different haplotypes, more recent haplotype analyses suggested that it occurred only once (PMID:29526279). This was an important lesson for the field. It would be great if the authors could connect their findings with the HbS story, and clearly state which beta-thalassemia mutations, if any, arose independently based on their data. I think the data is largely consistent with a single occurrence for most (all?) beta-thalassemia alleles that were studied.
4. I don't think that the data presented in SFig. 17 supports a causal role for rs72872549 since the ChIPseq data came from different experiments and cell-lines. Was the data normalized in the same way? What are the genotypes at rs72872549 in Hudep2 and K562? Why are they 2 tracks for Hudep2 in the figure? I think that the interpretation of this result needs to be toned down quite a bit.

(Remarks on code availability)

Not applicable.

Reviewer #2

(Remarks to the Author)

Review Summary

This manuscript presents a comprehensive investigation into the haplotype structure and evolutionary trajectories of β -thalassemia mutations in southern Chinese populations. Utilizing an extensive dataset—spanning over 20,000 individuals and comparative genomic resources—the authors identify three primary haplogroups at the β -globin locus and investigate the ancestral origins of several pathogenic variants. The study employs a broad range of analytical approaches, including haplotype network modeling, evolutionary dating, and association studies linking genetic variation to clinical and hematological phenotypes.

The paper addresses a significant gap in the literature, particularly given the high incidence of β -thalassemia in China and the limited understanding of regional haplotype variation. By combining evolutionary genetics with clinical relevance, the study makes a timely and valuable contribution. Overall, the manuscript is well-structured, clearly presented, and methodologically robust, though certain aspects would benefit from further clarification and expanded discussion.

1. Noteworthy Results

This study presents a large-scale and integrative analysis of β -thalassemia mutations in over 20,000 individuals from southern China. The most notable findings include:

- Identification of three primary haplogroups (HG1, HG2, HG3) at the β -globin locus, with HG1 showing unique expansion in East Asians.
- Evidence for recent and region-specific origins of several β -thalassemia mutations, particularly the -28, CD41/42, and -50 variants.
- Demonstration of haplotype-sharing patterns shaped by historical gene flow, admixture, and possible founder effects in geographically or ethnically distinct populations.
- The finding that HG2 is associated with elevated HbF levels and may modulate disease severity through interaction with the KLF1 transcription factor.

These results are valuable both in understanding the evolutionary history of hemoglobinopathies and for their potential clinical implications

2. Significance to the Field and Related Fields

The work holds clear significance in multiple domains such as Medical Genetics, Population Genomics, and Anthropology and Human Evolution.

3. Comparison to Established Literature

This study builds meaningfully on previous work (e.g., Shang et al., EBioMedicine 2017; Zhang et al., J Mol Evol 2008) by combining genomic, clinical, and evolutionary analyses across a much larger and more ethnically diverse cohort. The integration of haplotype analysis, allele age estimation, and phenotypic association within a single study is a notable advancement. Its approach to linking haplotype diversity with human migration and adaptation is comparable in style and ambition to studies in African or Mediterranean populations but fills a critical gap in East Asian data.

4. Support for Conclusions and Claims

The manuscript's major claims—regarding haplotype diversity, the regional origins of mutations, and modifier effects on phenotype—are generally well supported by the data. However, some interpretations, especially those linking HbF levels and malaria-related selection, would benefit from a more cautious framing. These hypotheses are plausible but could be presented as preliminary until functionally validated.

5. Flaws in Data Analysis, Interpretation, or Conclusions

No major methodological flaws are apparent. That said, a few concerns should be addressed:

- Causal Inference: Claims regarding selective pressures on HG2 and the functional consequences of the rs72872549-KLF1 interaction, while intriguing, are speculative without experimental validation.
- Gene Flow Interpretation: The paper could strengthen its conclusions about admixture by including more formal statistical tests (e.g., f -statistics or admixture graph modeling).
- Clarity in Figures: Some figures (e.g., Fig. 2 and 3) are quite dense and would benefit from visual simplification or more explicit highlighting of key patterns.

These do not prohibit publication but do warrant revision.

6. Revision or Rejection?

The manuscript merits major revision, not rejection. The core contributions are strong, but the interpretative sections would benefit from additional refining, and several figures could be improved for clarity.

7. Methodological Soundness

The methodologies used such as linkage disequilibrium mapping, haplotype clustering, TMRCA estimation, and SNP phenotype associations are standard, appropriate, and executed at scale. The use of deep sequencing data with high coverage strengthens the reliability of the genotyping and haplotype resolution.

8. Field Standards and Expectations

Yes, the study meets the expected scientific standards in the field. The scale, analytical depth, and integration of multiple data types reflect the scope typically expected for high-impact publications in genomics and evolutionary medicine.

9. Reproducibility of Methods

The methods section is generally thorough, with detailed information on the software, parameters, and datasets used. The inclusion of publicly available data repositories and links to scripts/tools also facilitates reproducibility. One improvement could be the addition of a schematic workflow diagram to help readers visualize the sequence of analytical steps.

Major Comments

1. Clarification of Sampling Strategy

While the manuscript emphasizes the distinctiveness of the southern Chinese context, a clearer justification for limiting the analysis to this region—rather than expanding to include broader Han or East Asian populations—would better situate the findings. A brief comparative note on β -thalassemia mutation profiles in northern China could further support the rationale.

2. Interpretation of Evolutionary Timing and Selection

The estimated ages of β -thalassemia mutations and their potential links to malaria selection are intriguing. However, the discussion would benefit from deeper engagement with archaeological and historical evidence regarding malaria in the region. Additionally, acknowledging the limitations of allele age estimation—such as the effects of population structure and selection—would improve transparency.

3. Support for Gene Flow Hypotheses

The proposed role of gene flow in shaping haplotype diversity in areas like Yunnan and Hainan is plausible, but the conclusions would be more robust if supported by additional analyses or references—such as f_3 statistics or admixture graph models—to substantiate these key claims.

4. Biological Significance of HG2

The association between the HG2 haplogroup and hematological traits, including its interaction with KLF1, is a compelling finding. However, the proposed connection to malaria-related selective pressures and elevated HbF levels should be framed with greater caution, as further experimental validation would be necessary to fully substantiate this hypothesis.

5. Figure Complexity and Accessibility

Figures 2 and 3 contain valuable data but are visually dense. Streamlining these visuals—by emphasizing key trends or using annotations such as arrows or summary boxes—would improve readability. Additionally, expanding the figure legends with more interpretive context could aid comprehension.

Minor Comments

- **Terminology Uniformity:** Please ensure consistent use of terms throughout the manuscript (e.g., HN-Li vs. HN_Li vs. HN Li) to avoid confusion, especially when referencing multiple population groups.
- **Use of Supplementary Material:** Since the manuscript frequently refers to supplementary figures and tables, consider incorporating one or two of the most central visuals directly into the main text to highlight key findings.
- **Typographical Corrections:** A few minor errors need correction (e.g., “demonstrats” on line 61 should be “demonstrates”).
- **Methodological Clarity:** A visual summary or flowchart outlining the analytical pipeline—from sequencing and phasing to haplotype analysis and phenotype associations—would greatly enhance the clarity of the methods section.

Recommendation

Major Revision

This is a high-quality manuscript with important implications for both evolutionary genetics and clinical genomics. I believe the paper has strong potential for publication in Nature Communications following revisions. The authors are encouraged to refine their interpretation of selection dynamics, strengthen their treatment of gene flow, and enhance figure clarity to make the work more accessible to a broader readership.

(Remarks on code availability)

Reviewer #3

(Remarks to the Author)

Zhang et al. describe the haplotype structure of the HBB locus in the south Chinese population (20,2022 individuals included), with emphasis on the distribution of beta-thalassemia variants (550 patients included). They extend this with hematological phenotyping which reveals an ameliorating effect of haplotype group 2 on beta-thalassemia. This study is well executed and the authors are to be congratulated on the clear presentation and careful preparation of the manuscript. I have a few minor comments:

It would be helpful to the reader if the consequences of main variants affecting HBB (e.g. amino acid change, frame shift, stop codon, splicing, regulatory (promoter)) were explained or included in Supplementary Table 3.

The authors predict that rs72872549 (G>C) resides within a KLF1 binding site. However, the highlighted bases (Supplementary Figure 17c) do not match the consensus very well. The location of the SNP, somewhere in the middle of the HBE gene, sits uncomfortable with a molecular mechanism explaining how KLF1 binding would lead to suppression of HBG1/2. The reanalysis of published ChIP-seq data in HUDEP2 and K562 cells (Supplementary Figure 17d) also does not support the hypothesis. If anything, this shows that KLF1 does not bind to this area in HUDEP2 cells. I suggest the authors either provide orthogonal experiments, for instance KLF1 EMSAs with oligonucleotides covering rs72872549 major and minor alleles, or leave this part out.

The Discussion section is interesting, thoughtful and well written. However, it is also very extensive (5 pages). In the interest of legibility the authors should consider reducing the length of this section.

(Remarks on code availability)

No new code was developed.

Version 1:

Reviewer comments:

Reviewer #1

(Remarks to the Author)
No additional comments.

(Remarks on code availability)

Reviewer #2

(Remarks to the Author)
Dear Authors,

Thank you for your response.
All my previous comments have been adequately addressed, and the revised manuscript reflects the necessary clarifications and improvements.
I have no further concerns.
I recommend this paper for acceptance.

(Remarks on code availability)

Reviewer #3

(Remarks to the Author)
The authors have addressed my queries satisfactorily.

(Remarks on code availability)

In cases where reviewers are anonymous, credit should be given to 'Anonymous Referee' and the source.
The images or other third party material in this Peer Review File are included in the article's Creative Commons license, unless indicated otherwise in a credit line to the material. If material is not included in the article's Creative Commons license and your intended use is not permitted by statutory regulation or exceeds the permitted use, you will need to obtain

Point-to-Point Response to the Reviewers' Comments

Please note that all the page and line numbers indicated in the responses are based on the clean.highlight version of the manuscript.

REVIEWER COMMENTS

Reviewer #1 (Remarks to the Author):

In this manuscript, Zhang and colleagues re-analyzed targeted next-generation DNA sequencing data of the beta-globin haplotype to infer the genetic history of beta-thalassemia mutations in Southern China. The beta-globin haplotype is interesting because it has been continuously under natural (positive and balancing) selection throughout human evolution. In China, like in other parts of the World where malaria is (or was) endemic, mutations arose that confer a protective advantage when heterozygous, but are detrimental in the homozygous states. In their study, the authors describe how allele, haplotype and haplogroups frequencies vary in Southern China, and proposed several hypotheses (e.g. founder effect in Hainan, the Silk Road for the HbE mutation) that may explain these fluctuations. Finally, they showed how the beta-globin haplotypes (and polymorphisms within) associate with hemoglobin levels in healthy participants and blood transfusion in beta-thalassemia patients.

This is a re-analysis of existing data. While the phylogeny analyses are robust (given the local genetic data available), the conclusions are not particularly novel. I am also not very convinced by their attempt to annotate in silico one of their top variants.

Response: We sincerely thank the reviewer for providing thoughtful feedback. We have thoroughly revised the manuscript based on the following comments and suggests. Although using previously published targeted sequencing data, the inclusion of a large population cohort analyzed in this study provided an unprecedented opportunity to investigate the genetic architecture and evolutionary dynamics of disease-associated mutations. We highlighted two key aspects of the novelty and significance of this work:

First, based on a systematic characterization of the β -globin locus in southern East Asia, a region with high disease prevalence and diverse β -thalassemia mutational profiles across

multiple ethnic populations, we observed heterogeneous haplotype backgrounds in different β -thalassemia mutations across populations. We found substantial genetic differentiations particularly between mainland China and Hainan Island, and proposed a multi-centric origin model of the β -thalassemia mutations in southern East Asia. Our findings suggest that lineage-specific natural selection and extensive gene flow within southern China and with neighboring regions further contributed to the observed β -thalassemia mutation diversity.

Second, we found a significant association between a specific haplotype group and hemoglobin indices in the southern Chinese natural populations, and further validated this clinical association in a β -thalassemia disease cohort. These findings suggest that haplotype background may represent a previously underappreciated determinant of clinical phenotypic variability in β -thalassemia.

Additional revisions, including clarification of the interpretation regarding the rs72872549–KLF1 interaction, are detailed below.

I have the following comments:

Comment 1.1 | I think that it should be made clear in the abstract and at the beginning of the Results section that this is not a whole-genome sequence project (this was my initial impression).

Response: We thank the reviewer for this helpful suggestion, which may help prevent potential misunderstandings. We have clarified in both the Abstract, Introduction and Results section that our analyses focus on the β -globin locus by targeted sequencing (Lines 107, 120).

Comment 1.2 | Given the lack of genome-wide genetic data, how did the authors ascertain ethnicity or genetic structure? Is that self-reported? How accurate are these self-reports? Would 5-10% inaccuracies (for instance for the residents of Hainan) impact your main conclusions?

Response: We appreciate the reviewer's comments and fully understand the concerns raised. In this study, ethnicity information was collected through standardized questionnaires and cross-verified using local household registries. While self-reported ethnicity may not capture individual-level ancestry with complete precision, it typically aligns well with population-level genetic structure. We assessed the population genetic structure using targeted sequencing data

spanning a total of 275.2 kb across chromosomes. We constructed a population phylogenetic tree based on pairwise population genetic distances as measured by pairwise F_{ST} . This analysis included the southern Chinese populations as well as global populations from the 1000 Genomes Project and the Human Genome Diversity Project. The southern Chinese populations generally clustered consistent with their geographic locations and ethnic affiliations, and with their counterparts from the public datasets (Supplementary Figure 3 in the revised manuscript).

We acknowledge that distinguishing individual genetic identity, especially among closely related ethnic groups, is inherently limited, even when using genome-wide data. We agree with the reviewer that a low level of misclassification is possible; however, such inaccuracies are unlikely to substantially impact our main conclusions. For example, the proposed island origin of the -50 mutation in haplogroup H4 remains robust, as the H4-linked -50 mutation is markedly enriched in Hainan relative to the mainland.

To address this point, we have clarified the methodology, incorporated relevant analyses and results, revised corresponding interpretations, and explicitly discussed the data limitations in the revised manuscript (Lines 120-124, 510-517).

Comment 1.3 | While it was originally reported that the sickle cell mutation (HbS) arose independently on 5 different haplotypes, more recent haplotype analyses suggested that it occurred only once (PMID:29526279). This was an important lesson for the field. It would be great if the authors could connect their findings with the HbS story, and clearly state which beta-thalassemia mutations, if any, arose independently based on their data. I think the data is largely consistent with a single occurrence for most (all?) beta-thalassemia alleles that were studied.

Response: We greatly appreciate the reviewer's constructive suggestion. As noted, previous studies have suggested a single origin of the sickle hemoglobin (HbS) mutation (β^S) in central Africa during late Pleistocene to early Holocene, followed by geographic spread and diversification associated to the Bantu expansions ^{1, 2}. In parallel, our analyses did not support multiple independent origins for each single β -thalassemia mutation among southern Chinese populations. This holds true even for the four mutations (e.g., CD41/42, -50, HbE, and CD43) with the highest haplotype diversity in our dataset. Nevertheless, the broader landscape encompassing variable β -thalassemia mutations reveals more complex scenarios than those

documented for the HbS mutation in Africa. For instance, based on the haplotype phylogeny and allele age estimation (as detailed in the manuscript), we proposed a mainland origin of most mutations, including the highly prevalent CD41/42, and a likely island origin of the -50 mutation in Hainan. These results collectively support a multi-centric origin model for the β -thalassemia mutations in southern East Asia. To better contextualize our finding, we followed the reviewer's suggestion and included a comparison between the proposed model for the β -thalassemia mutations in southern East Asia and that for the HbS mutation in Africa in the revised Fig. 6.

Comment 1.4 | I don't think that the data presented in SFig. 17 supports a causal role for rs72872549 since the ChIPseq data came from different experiments and cell-lines. Was the data normalized in the same way? What are the genotypes at rs72872549 in Hudep2 and K562? Why are they 2 tracks for Hudep2 in the figure? I think that the interpretation of this result needs to be toned down quite a bit.

Response: We thank the reviewer for raising this important issue. We fully agree that the ChIP-seq data presented in the previous Supplementary Figure 17 lacked sufficient interpretive power due to technical limitations. Specifically, the ChIP-seq tracks presented in our previous submission were from different experiments (K562 with one track vs. HUDEP-2 with two biological replicates). It was a pity that the raw sequencing files were not publicly available, which precluded unified normalization. In response to the reviewer's question on genotypes, we performed Sanger sequencing of rs72872549 in both K562 and HUDEP-2 cells. The results confirmed that K562 cells are homozygous for the G allele (GG), while HUDEP-2 cells are heterozygous (GC) at this locus. This finding is inconsistent with our initial hypothesis and reinforces the need for caution in interpreting KLF1 binding at this site. Accordingly, we have revised the manuscript to present a more cautious interpretation of rs72872549, acknowledging the need for further functional validation (Lines 431-445). We have also removed the ChIP-seq evidence entirely (previous Supplementary Figure 17d).

Reviewer #1 (Remarks on code availability):

Not applicable.

Reviewer #2 (Remarks to the Author):

Review Summary

This manuscript presents a comprehensive investigation into the haplotype structure and evolutionary trajectories of β -thalassemia mutations in southern Chinese populations. Utilizing an extensive dataset—spanning over 20,000 individuals and comparative genomic resources—the authors identify three primary haplogroups at the β -globin locus and investigate the ancestral origins of several pathogenic variants. The study employs a broad range of analytical approaches, including haplotype network modeling, evolutionary dating, and association studies linking genetic variation to clinical and hematological phenotypes.

The paper addresses a significant gap in the literature, particularly given the high incidence of β -thalassemia in China and the limited understanding of regional haplotype variation. By combining evolutionary genetics with clinical relevance, the study makes a timely and valuable contribution. Overall, the manuscript is well-structured, clearly presented, and methodologically robust, though certain aspects would benefit from further clarification and expanded discussion.

1. Noteworthy Results

This study presents a large-scale and integrative analysis of β -thalassemia mutations in over 20,000 individuals from southern China. The most notable findings include:

- Identification of three primary haplogroups (HG1, HG2, HG3) at the β -globin locus, with HG1 showing unique expansion in East Asians.
- Evidence for recent and region-specific origins of several β -thalassemia mutations, particularly the -28, CD41/42, and -50 variants.
- Demonstration of haplotype-sharing patterns shaped by historical gene flow, admixture, and possible founder effects in geographically or ethnically distinct populations.
- The finding that HG2 is associated with elevated HbF levels and may modulate disease severity through interaction with the KLF1 transcription factor.

These results are valuable both in understanding the evolutionary history of hemoglobinopathies and for their potential clinical implications.

2. Significance to the Field and Related Fields

The work holds clear significance in multiple domains such as Medical Genetics, Population Genomics, and Anthropology and Human Evolution.

3. Comparison to Established Literature

This study builds meaningfully on previous work (e.g., Shang et al., *EbioMedicine* 2017; Zhang et al., *J Mol Evol* 2008) by combining genomic, clinical, and evolutionary analyses across a much larger and more ethnically diverse cohort. The integration of haplotype analysis, allele age estimation, and phenotypic association within a single study is a notable advancement. Its approach to linking haplotype diversity with human migration and adaptation is comparable in style and ambition to studies in African or Mediterranean populations but fills a critical gap in East Asian data.

4. Support for Conclusions and Claims

The manuscript's major claims—regarding haplotype diversity, the regional origins of mutations, and modifier effects on phenotype—are generally well supported by the data. However, some interpretations, especially those linking HbF levels and malaria-related selection, would benefit from a more cautious framing. These hypotheses are plausible but could be presented as preliminary until functionally validated.

5. Flaws in Data Analysis, Interpretation, or Conclusions

No major methodological flaws are apparent. That said, a few concerns should be addressed:

- **Causal Inference:** Claims regarding selective pressures on HG2 and the functional consequences of the rs72872549-KLF1 interaction, while intriguing, are speculative without experimental validation.

- Gene Flow Interpretation: The paper could strengthen its conclusions about admixture by including more formal statistical tests (e.g., f-statistics or admixture graph modeling).
- Clarity in Figures: Some figures (e.g., Fig. 2 and 3) are quite dense and would benefit from visual simplification or more explicit highlighting of key patterns.

These do not prohibit publication but do warrant revision.

6. Revision or Rejection?

The manuscript merits major revision, not rejection. The core contributions are strong, but the interpretative sections would benefit from additional refining, and several figures could be improved for clarity.

7. Methodological Soundness

The methodologies used such as linkage disequilibrium mapping, haplotype clustering, TMRCA estimation, and SNP phenotype associations are standard, appropriate, and executed at scale. The use of deep sequencing data with high coverage strengthens the reliability of the genotyping and haplotype resolution.

8. Field Standards and Expectations

Yes, the study meets the expected scientific standards in the field. The scale, analytical depth, and integration of multiple data types reflect the scope typically expected for high-impact publications in genomics and evolutionary medicine.

9. Reproducibility of Methods

The methods section is generally thorough, with detailed information on the software, parameters, and datasets used. The inclusion of publicly available data repositories and links to scripts/tools also facilitates reproducibility. One improvement could be the addition of a schematic workflow diagram to help readers visualize the sequence of analytical steps.

Response: We sincerely thank the reviewer for the positive and constructive evaluation of our work. We greatly appreciate the recognition of the significance, methodological rigor, and clarity

of our manuscript. We have carefully considered all comments and suggestions and have revised the manuscript accordingly to improve clarity and depth of discussion. Detailed responses to each point are provided below.

Major Comments

Comment 2.1 | Clarification of Sampling Strategy

While the manuscript emphasizes the distinctiveness of the southern Chinese context, a clearer justification for limiting the analysis to this region—rather than expanding to include broader Han or East Asian populations—would better situate the findings. A brief comparative note on β -thalassemia mutation profiles in northern China could further support the rationale.

Response: We thank the reviewer for this insightful comment. We focused on southern China due to its high prevalence and diversity of β -thalassemia mutations. Thalassemia exhibits significant regional disparity across China, with a pronounced north–south gradient in prevalence (Figure R1)³. In particular, the incidence of thalassemia in northern China was extremely low prior to the industrialization and large-scale internal migration in China within the past two decades^{4, 5}. Limited studies of sporadic cases identified in the north, particularly among individuals with ancestral roots in that region, have reported IVS-II-654, CD41/42, and CD17 as the most common mutations. This mutation spectrum closely resembles that which we observed in southern Chinese populations. The southern concentration of β -thalassemia is shaped by both historical and biological factors, including past malaria endemicity, which exerted selective pressure favoring β -thalassemia alleles. Our goal is to trace the genetic and evolutionary basis of the β -thalassemia mutational complexity in this area where the clinical and genetic burden of the disease is most significant.

We have included comparative population genetic analyses using the global populations released by the public datasets, with a particular focus on the East and Southeast Asian populations, as detailed in Supplementary Table 2 and Supplementary Figure 6 of the revised manuscript. Although the β -thalassemia mutational spectrum in Southeast Asia also differs from that in southern China (Figure R2)⁵, these comparisons offer valuable insights into how population admixture and gene flow have shaped the genetic architecture of the disease.

Together, these analyses help contextualize our findings within the broader Asian genomic landscape.

In the revised manuscript, we have incorporated a clarifying statement in the Introduction section to inform readers about the paucity of β -thalassemia in northern China (Lines 71-73).

[Editorial note: figure redacted]

Figure R1. Prevalence of β -thalassemia in different regions of mainland China ³

Figure R2. Comparison of β -thalassemia mutation spectrum between China and Southeast Asia ⁵.

Comment 2.2 | Interpretation of Evolutionary Timing and Selection

The estimated ages of β -thalassemia mutations and their potential links to malaria selection are intriguing. However, the discussion would benefit from deeper engagement with archaeological and historical evidence regarding malaria in the region. Additionally, acknowledging the limitations of allele age estimation—such as the effects of population structure and selection—would improve transparency.

Response: We thank the reviewer for this insightful suggestion. To strengthen the discussion of the evolutionary history of β -thalassemia mutations and their potential connection to malaria-driven selection, we have expanded the Discussion section to incorporate relevant archaeological and historical evidence concerning the spread of malaria in southern East Asia (Lines 337-355). Additionally, we included supporting evidence of population migrations and gene flow to provide greater context and improve transparency. These insights have also been integrated into the revised Fig. 6. As suggested, we have discussed the limitation of allele age estimation in the revised manuscript (Lines 453-461).

Comment 2.3 | Support for Gene Flow Hypotheses

The proposed role of gene flow in shaping haplotype diversity in areas like Yunnan and Hainan is plausible, but the conclusions would be more robust if supported by additional analyses or references—such as f_3 statistics or admixture graph models—to substantiate these key claims.

Response: We thank the reviewer for this thoughtful suggestion. We have strengthened our conclusions by referencing genome-wide studies on gene flow in both Yunnan and Hainan, integrating genetic and historical evidence (Lines 350-360, 393-410) These references offer a robust foundation for our interpretations:

Despite its linguistic and cultural diversity, most Hainan populations trace their ancestries to southern mainland China ⁶. The population is primarily composed of the Han Chinese (80%) and Li (20%). The HN-Li people, descended from the ancient Bai-Yue, were thought to be the earliest settlers and have been residing on the island for over 4,000 years ⁷. The Han Chinese migration to Hainan began during the Qin Dynasty (~2,200 years ago), with major influxes in the 16th and 17th centuries from Fujian and Guangdong ⁸.

Yunnan has long served as a crossroads of historic trade, cultural exchange, and human migration, contributing significantly to both regional and global history. The Southern Silk Road, starting around 2,000 years ago, connected central China with mainland Southeast Asia, eventually linking to the Maritime Silk Road and extending routes to India and beyond ⁹. Additionally, Yunnan was central to the Ancient Tea Horse Road, a trade network that began approximately 1,500 years ago and linked the tea-producing regions of southwest Yunnan with

markets in Tibet, Southeast Asia, Nepal, and central China¹⁰. Genetic studies have revealed shared ancestries and extensive gene flow between southern China, mainland Southeast Asia, and South Asia^{11, 12, 13, 14}. In particular, recent ancient genome evidence suggests that Yunnan may have been a source region for the Austroasiatic expansion into Southeast Asia and northeast India within the past 5,500 years, which coincides with the period of historically reported malaria endemicity in China¹⁵.

Together, these genetic and historical lines of evidence provide strong support for the role of gene flow in shaping the haplotype diversity of β -thalassemia mutations in these regions. We have added this evidence into the revised manuscript (Lines 350-360, 393-410).

Comment 2.4 | Biological Significance of HG2

The association between the HG2 haplogroup and hematological traits, including its interaction with KLF1, is a compelling finding. However, the proposed connection to malaria-related selective pressures and elevated HbF levels should be framed with greater caution, as further experimental validation would be necessary to fully substantiate this hypothesis.

Response: We sincerely appreciate the reviewer's insightful comments regarding the interpretation of the biological significance of the HG2 haplogroup. As suggested, we have now revised our manuscript to explicitly acknowledge the necessity of caution in drawing connections between HG2, elevated HbF levels, and selective pressures exerted by malaria. Although the current epidemiological data and previous literature indicate potential evolutionary implications, we fully recognize that these hypotheses remain preliminary without solid, extensive experimental validation. We have revised relevant section of the manuscript to incorporate the reviewer's suggestions and improve clarity (Lines 431-445).

Comment 2.5 | Figure Complexity and Accessibility

Figures 2 and 3 contain valuable data but are visually dense. Streamlining these visuals—by emphasizing key trends or using annotations such as arrows or summary boxes—would improve readability. Additionally, expanding the figure legends with more interpretive context could aid comprehension.

Response: We thank the reviewer for this helpful suggestion. We have substantially revised all the main figures (including Figure 2 and Figure 3 as mentioned by the reviewer) to enhance clarity and visual presentation. In addition, we have expanded the figure legends to provide more interpretive context, aiding comprehension and better guiding the reader through the data.

Minor Comments

Comment 2.6 | Terminology Uniformity: Please ensure consistent use of terms throughout the manuscript (e.g., HN-Li vs. HN_Li vs. HN Li) to avoid confusion, especially when referencing multiple population groups.

Response: We appreciate the reviewer’s attention to detail. We have carefully reviewed and revised the manuscript to ensure consistent use of population group labels. All instances now follow a standardized format using hyphens (e.g., HN-Li) throughout the main text and Supplementary Materials.

Comment 2.7 | Use of Supplementary Material: Since the manuscript frequently refers to supplementary figures and tables, consider incorporating one or two of the most central visuals directly into the main text to highlight key findings.

Response: We have re-organized the figures and tables throughout the manuscript in accordance with the reviewer’s suggestion.

Comment 2.8 | Typographical Corrections: A few minor errors need correction (e.g., “demonstrats” on line 61 should be “demonstrates”).

Response: We thank the reviewer for pointing out the apparent typographical errors. We have conducted additional rounds of proofreading to eliminate these issues.

Comment 2.9 | Methodological Clarity: A visual summary or flowchart outlining the analytical pipeline—from sequencing and phasing to haplotype analysis and phenotype associations—would greatly enhance the clarity of the methods section.

Response: We thank the reviewer for this constructive suggestion. To improve clarity, we have included a workflow chart in Supplementary Figure 2 of the revised manuscript.

Recommendation

Major Revision

This is a high-quality manuscript with important implications for both evolutionary genetics and clinical genomics. I believe the paper has strong potential for publication in Nature Communications following revisions. The authors are encouraged to refine their interpretation of selection dynamics, strengthen their treatment of gene flow, and enhance figure clarity to make the work more accessible to a broader readership.

Response: We sincerely thank the reviewer for the positive assessment and valuable feedback. We have enhanced the interpretation of our results and improved the presentation throughout the study to make the work more accessible to a broader readership.

Reviewer #3 (Remarks to the Author):

Zhang et al. describe the haplotype structure of the HBB locus in the south Chinese population (20,2022 individuals included), with emphasis on the distribution of beta-thalassemia variants (550 patients included). They extend this with hematological phenotyping which reveals an ameliorating effect of haplotype group 2 on beta-thalassemia. This study is well executed and the authors are to be congratulated on the clear presentation and careful preparation of the manuscript.

Response: We sincerely thank the reviewer for the positive and encouraging feedback. We have carefully considered all the following comments and incorporated revisions to further improve the manuscript. Detailed responses to individual points are provided below.

I have a few minor comments:

Comment 3.1 | It would be helpful to the reader if the consequences of main variants affecting HBB (e.g. amino acid change, frame shift, stop codon, splicing, regulatory (promoter)) were explained or included in Supplementary Table 3.

Response: We agree with the reviewer that including functional annotations would enhance clarity and aid interpretation. Accordingly, we have included an additional column in

Supplementary Table 5 of the revised manuscript (corresponding to Supplementary Table 3 in the original version), which describes the predicted molecular consequence of each variant.

Comment 3.2 | The authors predict that rs72872549 (G>C) resides within a KLF1 binding site. However, the highlighted bases (Supplementary Figure 17c) do not match the consensus very well. The location of the SNP, somewhere in the middle of the HBE gene, sits uncomfortable with a molecular mechanism explaining how KLF1 binding would lead to suppression of HBG1/2. The reanalysis of published ChIP-seq data in HUDEP2 and K562 cells (Supplementary Figure 17d) also does not support the hypothesis. If anything, this shows that KLF1 does not bind to this area in HUDEP2 cells. I suggest the authors either provide orthogonal experiments, for instance KLF1 EMSAs with oligonucleotides covering rs72872549 major and minor alleles, or leave this part out.

Response: We thank the reviewer for the thoughtful and constructive comments. We acknowledge the limitations in our original interpretation and have removed speculative statements regarding the potential role of rs72872549 in KLF1 binding. In response to the reviewer's concern, we conducted additional motif scanning using the JASPAR 2024 database (motif ID: MA0493.2). This analysis identified two weakly scoring candidate KLF1 motifs flanking rs72872549 for the reference G allele (raw scores: 7.73 and 9.1; relative scores: 0.88 and 0.90). Both matches were abolished by the alternative C allele. Although these predicted motifs exceed the default relative score threshold (0.80), they fall short of the high-confidence threshold (e.g., >0.95) typically applied in functional studies (e.g., Cheng et al., Nat Genet 2021), leaving the binding potential speculative.

Furthermore, rs72872549 is located outside known regulatory region. Our reanalysis of publicly available ChIP-seq data from HUDEP-2 and K562 cells (previously shown in Supplementary Fig. 17d) did not provide compelling evidence of KLF1 occupancy at this site. However, given the limitations of these datasets, including limited metadata and restricted raw reads files, cross-experiment normalization and direct comparisons are not feasible.

We agree with the reviewer that the current evidence does not support a mechanistic link between rs72872549 and KLF1 binding. We have therefore removed Supplementary Figure 17d and revised Supplementary Figure 17c to present only the quantitative motif scan, now

summarized in Supplementary Table 10). Speculative statements regarding this variant have been removed from the main text, and our interpretation has been reframed more cautiously.

As suggested, functional assays, such as EMSA or reporter-based analyses, would be essential to rigorously test allele-specific effects on transcription factor binding, but these experiments are beyond the scope of the current haplotype-based population study. We hope these revisions provide a more measured interpretation and address the reviewer's concerns. The relevant paragraph in the manuscript has been updated accordingly (Lines 431-445).

Comment 3.3 | The Discussion section is interesting, thoughtful and well written. However, it is also very extensive (5 pages). In the interest of legibility the authors should consider reducing the length of this section.

Response: We thank the reviewer for the positive and encouraging feedback. We have revised the Discussion section to enhance clarity and conciseness.

Reviewer #3 (Remarks on code availability):

No new code was developed.

Response: We have uploaded the code in Code Ocean capsule, which is available at <https://codeocean.com/capsule/9987989/tree>.

References

1. Shriner D, Rotimi CN. Whole-Genome-Sequence-Based Haplotypes Reveal Single Origin of the Sickle Allele during the Holocene Wet Phase. *Am J Hum Genet* **102**, 547-556 (2018).
2. Laval G, *et al.* Recent Adaptive Acquisition by African Rainforest Hunter-Gatherers of the Late Pleistocene Sickle-Cell Mutation Suggests Past Differences in Malaria Exposure. *Am J Hum Genet* **104**, 553-561 (2019).
3. Lai K, Huang G, Su L, He Y. The prevalence of thalassemia in mainland China: evidence from epidemiological surveys. *Sci Rep* **7**, 920 (2017).
4. Yang Z, Zhou W, Cui Q, Qiu L, Han B. Gene spectrum analysis of thalassemia for people residing in northern China. *BMC Med Genet* **20**, 86 (2019).
5. Yang Z, Cui Q, Zhou W, Qiu L, Han B. Comparison of gene mutation spectrum of thalassemia in different regions of China and Southeast Asia. *Mol Genet Genomic Med* **7**, e680 (2019).
6. Peng MS, He JD, Liu HX, Zhang YP. Tracing the legacy of the early Hainan Islanders--a perspective from mitochondrial DNA. *BMC Evol Biol* **11**, 46 (2011).
7. Chen H, *et al.* Tracing Bai-Yue Ancestry in Aboriginal Li People on Hainan Island. *Mol Biol Evol* **39**, (2022).
8. He G, *et al.* Inferring the population history of Tai-Kadai-speaking people and southernmost Han Chinese on Hainan Island by genome-wide array genotyping. *Eur J Hum Genet* **28**, 1111-1123 (2020).
9. Yang B. *Between Winds and Clouds: The Making of Yunnan (second century BCE to twentieth century CE)*. Columbia University Press (2014).
10. Sigley G. Cultural Heritage Tourism and the Ancient Tea Horse Road of Southwest China. *International Journal of China Studies* **1**, (2010).
11. Bai F, *et al.* Paleolithic genetic link between Southern China and Mainland Southeast Asia revealed by ancient mitochondrial genomes. *Journal of Human Genetics* **65**, 1125-1128 (2020).
12. Consortium HP-AS, *et al.* Mapping human genetic diversity in Asia. *Science* **326**, 1541-1545 (2009).
13. Chandrasekar A, *et al.* Updating phylogeny of mitochondrial DNA macrohaplogroup m in India: dispersal of modern human in South Asian corridor. *PLoS One* **4**, e7447 (2009).
14. Wei X, *et al.* Neolithic to Bronze Age human maternal genetic history in Yunnan, China. *J Genet Genomics* **52**, 483-493 (2025).
15. Wang T, *et al.* Prehistoric genomes from Yunnan reveal ancestry related to Tibetans and Austroasiatic speakers. *Science* **388**, eadq9792 (2025).